# ImAD: An End-to-End Method for Unsupervised Anomaly Detection in the Presence of Missing Values

## Abstract

Common anomaly detection methods require fully observed data for model training and inference and cannot handle data containing missing values. The missing data problem is pervasive in various real-world scenarios but the study of anomaly detection with missing data is quite limited. In this work, we first construct and evaluate a straightforward strategy, "impute-then-detect", which combines state-of-the-art data imputation methods with unsupervised anomaly detection methods, where the training data are only composed of normal samples. We observe that such two-stage methods often yield imputation bias for normal data, namely, the imputation methods are inclined to make incomplete samples "normal". The fundamental reason is that the imputation models are learned from normal data and cannot be generalized to abnormal data. To solve the challenging problem, we propose an end-to-end method called ImAD for unsupervised anomaly detection in the presence of missing values. ImAD integrates data imputation with anomaly detection into a unified optimization problem and introduces well-designed pseudo-abnormal samples to ensure the discrimination ability of the imputation process. Experiments in the settings of three different missing mechanisms show that the proposed ImAD alleviates the imputation bias and achieves much better detection performance in comparison to the baselines.

## 1 Introduction

Anomaly detection (AD) (Breunig et al., 2000; Schölkopf et al., 2001; Liu et al., 2008; Pevnỳ, 2016; Zong et al., 2018; Ruff et al., 2018; Cai & Fan, 2022), aiming at identifying anomalous samples in data, is a crucial machine learning problem and is widely applied in many high-stakes fields such as healthcare, finance, and cybersecurity. Existing anomaly detection methods commonly require that both the training and test sets are composed of complete data and they cannot handle data containing missing values. Data missing is a long-standing and unavoidable problem in many real-world scenarios such as healthcare and finance. Commonly, the collection, transmission, and storage process of data may cause missing values and yield incomplete data. It is necessary and inevitable to solve the anomaly detection problem in the presence of missing values. A naive strategy is to fill in the missing values by statistical characteristics such as mean, median, or mode and then perform anomaly detection. Taking two real-world tabular datasets "Adult" and "KDD" datasets as examples, we use variable means to fill the missing entries and then conduct two classical AD methods (OC-SVM (Schölkopf et al., 2001) and Isolation Forest (Liu et al., 2008)) and two deep learning based AD methods (Deep SVDD (Ruff et al., 2018) and NeutraL AD (Qiu et al., 2021)). The results are shown in Figure 1. The detection performances of the four anomaly detection methods degrade significantly when the missing rate increases. This result verified the failure of the naive strategy and the difficulty of unsupervised anomaly detection in the presence of missing values.

Besides the aforementioned naive approach, one may consider using more powerful missing imputation algorithms to recover the missing values. Indeed, in the past decades, a substantial amount of research has been dedicated to developing missing data imputation algorithms (Wilks, 1932; Ghahramani & Jordan, 1993; Dempster et al., 1977; Pigott, 2001; Candes & Recht, 2012; Stekhoven & Bühlmann, 2012; Fan & Chow, 2017; Gondara & Wang, 2018; Yoon et al., 2018; Fan et al., 2020; Muzellec et al., 2020).

However, in the framework of unsupervised anomaly detection, where the training data are composed of normal samples, such "impute-then-detect" pipeline method would yield imputation bias for normal data, i.e., the imputation methods are inclined to recover an abnormal sample with missing values as "normal" as possible. The reason is that the training set and testing set do not satisfy the condition of identical distribution and the learned imputation model does not generalize well. The imputation bias significantly lowers the accuracy of anomaly detection. In Section 4.3, we quantitatively evaluate the "impute-then-detect" pipeline using state-of-the-art imputation algorithms and AD algorithms.

In this work, we propose a novel method, called ImAD, for unsupervised anomaly detection on data with missing values. The main idea of

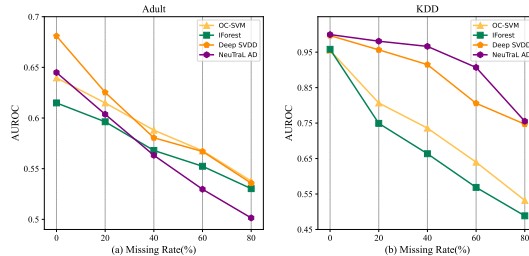

Figure 1: Performance degradation of anomaly detection methods with the increasing missing rate. Plots (a) and (b) correspond to the "Adult" and "KDD" datasets, respectively. The missing rate denotes the missing probability of each entry under MCAR (missing completely at random), and AUROC is the Area Under the Receiver Operating Characteristic curve.

ImAD is to integrate data imputation and anomaly detection into a unified optimization objective and alleviate imputation bias by automatically generating some pseudo-abnormal samples. Note that the pseudo-abnormal samples are from the training process and we do not use any extra data in our experiments. ImAD consists of three modules including an imputation module, a projection module, and a reconstruction module. The imputation module is used to recover missing values, the projection module maps imputed data into compact and bounded target distributions, and the reconstruction module aims to preserve key information from the original data distribution. Our contributions are summarized as follows.

- We study the imputation bias problem of the "impute-then-detect" pipeline and quantitatively evaluate their detection performance.

- We propose a novel method ImAD for anomaly detection on incomplete data. To the best of our knowledge, ImAD is the first end-to-end unsupervised anomaly detection method for incomplete data. We also provide theoretical analysis for the effectiveness of ImAD.

- We evaluate the detection performance of all baselines on seven real-world datasets, considering three different missing mechanisms (missing completely at random (MCAR), missing at random (MAR) and missing not at random (MNAR)).

## 2 RELATED WORK

### 2.1 MISSING DATA IMPUTATION

The goal of data imputation is to fill missing entries of data with plausible values and provide the imputed data for downstream tasks such as classification, clustering, and visualization. As the missing data problem is prevalent in many fields, the study on missing data imputation is extensive and many algorithms have been proposed in the past decades. Mayer et al. (2019) pointed out that there are approximately 150 implementations available to handle missing data. These methods can be roughly organized into three categories. The first category is based on the iterative regression model, in which one of the most well-known methods is the Multiple Imputation by Chained Equations (MICE) (Royston & White, 2011). Stekhoven & Bühlmann (2012) presented MissForest by training random forests on observed data through an iterative imputation scheme. MissForest is very effective in categorical data imputation. The second category is the matrix completion methods (Candes & Recht, 2012; Mazumder et al., 2010; Fan et al., 2019; 2020). The third category is based on deep learning especially deep generative models (Fan & Chow, 2017; Yoon et al., 2018; Li et al., 2019; Muzellec et al., 2020). For instance, Yoon et al. (2018) proposed generative adversarial imputation network (GAIN) based on vanilla generative adversarial network (GAN) (Goodfellow et al., 2014) and Tashiro et al. (2021) proposed conditional score-based diffusion models for probabilistic time-series imputation (CSDI) based diffusion model (Sohl-Dickstein et al., 2015). Indeed, these

deep learning based imputation methods often achieve state-of-the-art performance in the tasks of missing data imputation, when the performance is often evaluated by the RMSE of imputation or the accuracy of a classifier when the data consists of multiple classes. However, their performance in recovering the missing values for anomaly detection is rarely studied.

## 2.2 Anomaly detection on incomplete data

The research on anomaly detection in the presence of missing values is very limited. To the best of the authors' knowledge, Zemicheal & Dietterich (2019) is the first work that evaluates the detection performance of anomaly detection methods combined with different data imputation techniques. Their experiments of anomaly detection with missing values on a few UCI datasets showed that implementations of unsupervised anomaly detection methods such as Isolation Forest (Liu et al., 2008) on incomplete data should always include algorithms for handling missing values. The imputation can significantly improve the performance of anomaly detection methods. Fan et al. (2022) studied the problem of statistical process monitoring with missing values and proposed a fast incremental nonlinear matrix completion method for online and sequential imputation. The imputation method is able to adapt to changes of data patterns. Sarda et al. (2023) provided a comparative study of seven unsupervised anomaly detection methods on GAN-imputed data.

It should be pointed out that the strategies used in (Zemicheal & Dietterich, 2019; Fan et al., 2022; Sarda et al., 2023) are two-stage methods, where the imputation models are trained on the training dataset that does not contain any abnormal data or only contains very few unlabeled outliers. As a result, the imputation model will not generalize well on abnormal data in the test stage and will use the learned pattern or structure of normal data to impute the missing values of abnormal data, which makes the abnormal data similar to normal data and hence lowers the detection accuracy. In contrast, in this work, we propose a novel end-to-end anomaly detection method in the presence of missing values. The imputation process of our method is able to preserve the normality or abnormality of the test data and hence ensures high detection accuracy.

## 3 Proposed Method

### 3.1 Problem Formulation and Our Motivation

Given $n$ samples $\mathbf{x}_1, \mathbf{x}_2, \cdots, \mathbf{x}_n$ drawn from an unknown distribution $\mathcal{D}_{\mathbf{x}} \subseteq \mathbb{R}^m$, the goal of unsupervised anomaly detection is to learn a decision function $f : \mathbb{R}^m \to \{0, 1\}$ by utilizing only these $n$ samples, such that $f(\mathbf{x}) = 0$ if $\mathbf{x} \in \mathcal{D}_{\mathbf{x}}$ and $f(\mathbf{x}) = 1$ if $\mathbf{x} \notin \mathcal{D}_{\mathbf{x}}$. We consider the scenario that $\mathbf{X} := [\mathbf{x}_1^\top, \mathbf{x}_2^\top, \cdots, \mathbf{x}_n^\top]^\top \in \mathbb{R}^{n \times m}$ contains missing values, which is often caused by the failure of data acquisition. For convenience, we let $\mathbf{M} \in \{0, 1\}^{n \times m}$ be a mask matrix determined by some missing mechanism $\mathcal{M}$ such as MCAR, MAR or MNAR, where $M_{i,j} = 1$ means $X_{i,j}$ is observed and $M_{i,j} = 0$ means $X_{i,j}$ is missing. Then, we observe the incomplete data matrix

$$\check{\mathbf{X}} = [\check{\mathbf{x}}_1^\top, \check{\mathbf{x}}_2^\top, \cdots, \check{\mathbf{x}}_n^\top]^\top = \mathbf{X} \odot \mathbf{M} \tag{1}$$

where $\odot$ is the Hadamard product. Equation (1) implies that the missing values of $\mathbf{X}$ are temporarily filled with zeros. As mentioned before, conventional anomaly detection methods are vulnerable to missing values and a good imputation algorithm can raise the detection accuracy of an anomaly detection method to some extent. However, the strategy "impute-then-detect" is inclined to make incomplete abnormal samples normal and hence cannot provide satisfactory detection performance.

In this work, we aim to provide an end-to-end anomaly detection method in the presence of missing values. The most challenging problem is that the imputation model (denoted as $\mathcal{I}$) trained only on normal data cannot generalize well to abnormal data. To solve this challenging problem, we propose to generate some pseudo-abnormal samples, and then learn an imputation model from both the original normal data and the generated pseudo-abnormal samples. Thus, the learned imputation model is able to generalize well to incomplete abnormal data in the test stage and recover the missing values with high accuracy, which further improves the accuracy of anomaly detection. Nevertheless, it is non-trivial to generate pseudo-abnormal samples because the distribution (i.e., $\mathcal{D}_{\mathbf{x}}$) of the training data is unknown and the dimension of the data $d$ is often high. We need to ensure that the generated pseudo-abnormal samples are similar enough to real abnormal data. On the other hand, the generated pseudo-abnormal samples should not be too far from the normal data, where a large gap will

make the learned imputation model fail to impute the abnormal samples close to normal data and cause the abnormal samples to be hard to detect.

## 3.2 Generating Pseudo-Abnormal Samples in Latent Space

Since $\mathcal{D}_{\mathbf{x}}$ is unknown and $m$ is often not small, it is very difficult to generate some meaningful pseudo-abnormal samples. Moreover, the incompleteness of $\mathbf{X}$ further increases the difficulty. Thus, we propose to find an $d$-dimensional latent space $\mathcal{Z}$ where the normal data are lying and then generate pseudo-abnormal samples around the normal samples in $\mathcal{Z}$. The samples in $\mathcal{Z}$ will be mapped back to the original data space. To be more precise, we define $\mathcal{D}_{\mathbf{z}}$ as the latent distribution of the normal data in $\mathcal{Z}$ and define $\mathcal{D}_{\tilde{\mathbf{z}}}$ as the latent distribution of pseudo-abnormal data in $\mathcal{Z}$. Since the patterns of normality are limited and the patterns of abnormality are unlimited, we let $\mathcal{D}_{\mathbf{z}}$ be a truncated Gaussian distribution in $\mathcal{Z}$ and assume that the remaining region of $\mathcal{Z}$ excluding the hyperball (denoted by $\mathcal{B}$, with radius $r_1$) defined by $\mathcal{D}_{\mathbf{z}}$ is the abnormal region, denoted as $\mathcal{Z} \setminus \mathcal{B}$. It should be pointed out that there is no need to define $\mathcal{D}_{\tilde{\mathbf{z}}}$ in the entire space $\mathcal{Z} \setminus \mathcal{B}$, which will be explained in the discussion for Theorem 3.1(b) in Section 3.6. Instead, we only need to define $\mathcal{D}_{\tilde{\mathbf{z}}}$ in a small region of $\mathcal{Z} \setminus \mathcal{B}$ that encloses $\mathcal{B}$, which will reduce the uncertainty of random sampling (or samples size equivalently) and make it easier for mapping the samples back to the original data space. Thus, we define $\mathcal{D}_{\tilde{\mathbf{z}}}$ as a hypershell surrounding $\mathcal{B}$ and let $\mathcal{D}_{\tilde{\mathbf{z}}}$ be a truncated Gaussian. The radii of two hyperspheres forming the hypershell are $r_1$ and $r_2$ respectively, where $r_2 > r_1$. An intuitive example is shown in Figure 2.

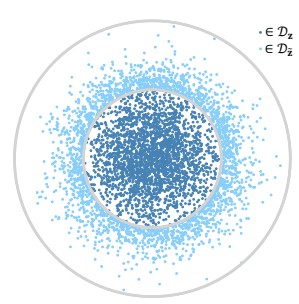

Figure 2: Visualization of $\mathcal{D}_{\mathbf{z}}$, and $\mathcal{D}_{\tilde{\mathbf{z}}}$ in a 2-D latent space $\mathcal{Z}$. $\mathcal{D}_{\mathbf{z}}$ and $\mathcal{D}_{\tilde{\mathbf{z}}}$ are truncated Gaussian from $\mathcal{N}(\mathbf{0}, 0.5^2 \cdot \mathbf{I}_2)$ and $\mathcal{N}(\mathbf{0}, \mathbf{I}_2)$ respectively. The radii of the inner and outer hyperspheres are $r_1$ and $r_2$ respectively.

## 3.3 Learning Framework

Given $\mathcal{D}_{\mathbf{z}}$, we learn a model $\mathcal{R} : \mathbb{R}^d \to \mathbb{R}^m$ to transform the samples drawn from $\mathcal{D}_{\mathbf{z}}$ to the original data distribution $\mathcal{D}_{\mathbf{x}}$, i.e.,

$$\mathcal{D}_{\mathbf{x}} \approx \mathcal{R}(\mathcal{D}_{\mathbf{z}}). \tag{2}$$

$\mathcal{R}$ is actually a reconstruction model that recovers the original data from the latent space $\mathcal{Z}$. With $\mathcal{D}_{\tilde{\mathbf{z}}}$ and $\mathcal{R}$, we can obtain a distribution of pseudo-abnormal data in the original data space as

$$\mathcal{D}_{\tilde{\mathbf{x}}} := \mathcal{R}(\mathcal{D}_{\tilde{\mathbf{z}}}). \tag{3}$$

The samples (denoted by $\tilde{\mathbf{x}}$) drawn from $\mathcal{D}_{\tilde{\mathbf{x}}}$ are reasonable pseudo-abnormal samples, which will be explained by the discussion for Theorem 3.1(a) in Section 3.6. Now we use a model $\mathcal{P} : \mathbb{R}^m \to \mathbb{R}^d$ to transform $\mathcal{D}_{\mathbf{x}}$ and $\mathcal{D}_{\tilde{\mathbf{x}}}$ into $\mathcal{D}_{\mathbf{z}}$ and $\mathcal{D}_{\tilde{\mathbf{z}}}$ respectively, i.e.,

$$\mathcal{D}_{\mathbf{z}} \approx \mathcal{P}(\mathcal{D}_{\mathbf{x}}), \quad \mathcal{D}_{\tilde{\mathbf{z}}} \approx \mathcal{P}(\mathcal{D}_{\tilde{\mathbf{x}}}). \tag{4}$$

However, $\mathbf{X}$ is incomplete, and we need to learn an imputation model $\mathcal{I}$ to recover the missing values, i.e., $\hat{\mathbf{X}} = \mathcal{I}(\check{\mathbf{X}})$. More generally, we denote

$$\mathcal{D}_{\hat{\mathbf{x}}} = \mathcal{I}(\mathcal{D}_{\check{\mathbf{x}}}). \tag{5}$$

We hope that the imputation model is also able to recover the missing values of the generated pseudo-abnormal samples if they have, though they are complete. We thus remove some values of the generated pseudo-abnormal samples $\tilde{\mathbf{x}} \sim \mathcal{D}_{\tilde{\mathbf{x}}}$ using some missing mechanism $\tilde{\mathcal{M}}$ and let $\mathcal{D}_{\check{\tilde{\mathbf{x}}}} = \tilde{\mathcal{M}}(\mathcal{D}_{\tilde{\mathbf{x}}})$. The missing values are then recovered by

$$\mathcal{D}_{\hat{\tilde{\mathbf{x}}}} = \mathcal{I}(\mathcal{D}_{\check{\tilde{\mathbf{x}}}}). \tag{6}$$

This addresses the problem of imputation bias encountered by the "impute-then-detect" methods.

Let $\mathcal{E}_I$, $\mathcal{E}_P$, and $\mathcal{E}_R$ denote some distance or discrepancy measure between distributions. We here show how to achieve the goals of (2), (3), (4), (5), and (6). First, for normal data, we solve

$$\underset{\mathcal{I},\mathcal{P},\mathcal{R}}{\text{minimize}} \; \mathcal{E}_I(\mathcal{I}(\mathcal{D}_{\check{\mathbf{x}}}), \mathcal{D}_{\tilde{\mathbf{x}}} \mid \mathcal{M}) + \mathcal{E}_P(\mathcal{P}(\mathcal{D}_{\hat{\mathbf{x}}}), \mathcal{D}_{\mathbf{z}}) + \mathcal{E}_R(\mathcal{R}(\mathcal{P}(\mathcal{D}_{\hat{\mathbf{x}}})), \mathcal{D}_{\check{\mathbf{x}}} \mid \mathcal{M}) \tag{7}$$

For the generated pseudo-abnormal data, we solve

$$\underset{\mathcal{I},\mathcal{P},\mathcal{R}}{\text{minimize}} \; \mathcal{E}_I\big(\mathcal{I}(\tilde{\mathcal{M}}(\mathcal{R}(\mathcal{D}_{\tilde{\mathbf{z}}}))), \tilde{\mathcal{M}}(\mathcal{R}(\mathcal{D}_{\tilde{\mathbf{z}}})) \mid \tilde{\mathcal{M}}\big) + \mathcal{E}_P\big(\mathcal{P}(\mathcal{I}(\tilde{\mathcal{M}}(\mathcal{R}(\mathcal{D}_{\tilde{\mathbf{z}}})))), \mathcal{D}_{\tilde{\mathbf{z}}}\big) \tag{8}$$

Let $\widehat{\mathcal{E}}$ be a finite-sample estimation of $\mathcal{E}$. Putting (7) and (8) together, we obtain the final formulation of our method ImAD as follows:

$$\underset{\mathcal{I},\mathcal{P},\mathcal{R}}{\text{minimize}} \; \underbrace{\widehat{\mathcal{E}}_I(\mathcal{I}([\check{\mathbf{X}};\check{\tilde{\mathbf{X}}}]), [\check{\mathbf{X}};\check{\tilde{\mathbf{X}}}] \mid [\mathbf{M},\tilde{\mathbf{M}}])}_{\mathcal{L}^{(\text{DI})}} + \underbrace{\widehat{\mathcal{E}}_P(\mathcal{P}([\hat{\mathbf{X}};\hat{\tilde{\mathbf{X}}}]), [\mathbf{Z};\tilde{\mathbf{Z}}])}_{\mathcal{L}^{(\text{AD})}} + \underbrace{\widehat{\mathcal{E}}_R(\mathcal{R}(\mathcal{P}(\hat{\mathbf{X}})), \check{\mathbf{X}} \mid \mathbf{M})}_{\mathcal{L}^{(\text{RE})}} \tag{9}$$

where $\check{\tilde{\mathbf{X}}} = \mathcal{R}(\tilde{\mathbf{Z}}) \odot \tilde{\mathbf{M}}$, $\hat{\tilde{\mathbf{X}}} = \mathcal{I}(\check{\tilde{\mathbf{X}}})$, and $[\cdot;\cdot]$ denotes the row-wise concatenation of two matrices. In (9), the samples in $\mathbf{Z}$ are drawn from $\mathcal{D}_{\mathbf{z}}$ and the samples in $\tilde{\mathbf{Z}}$ are drawn from $\mathcal{D}_{\tilde{\mathbf{z}}}$. The roles of the three parts of the objective function in (9) are analyzed as follows.

- $\mathcal{L}^{(\text{DI})}$ denotes the data imputation loss. With this loss, the imputation model will be able to recover the missing values of normal data and abnormal data.
- $\mathcal{L}^{(\text{AD})}$ denotes the anomaly detection loss. With this loss, the anomaly detection model will be discriminative and be able to project normal data and abnormal data into different regions in $\mathcal{Z}$.
- $\mathcal{L}^{(\text{RE})}$ denotes the reconstruction loss. This loss is to ensure that $\mathcal{D}_{\mathbf{z}}$ and $\mathcal{D}_{\tilde{\mathbf{z}}}$ are meaningful.

We see that our method ImAD couples data imputation with anomaly detection to a unified optimization objective. Figure 3 depicts the overall architecture of ImAD.

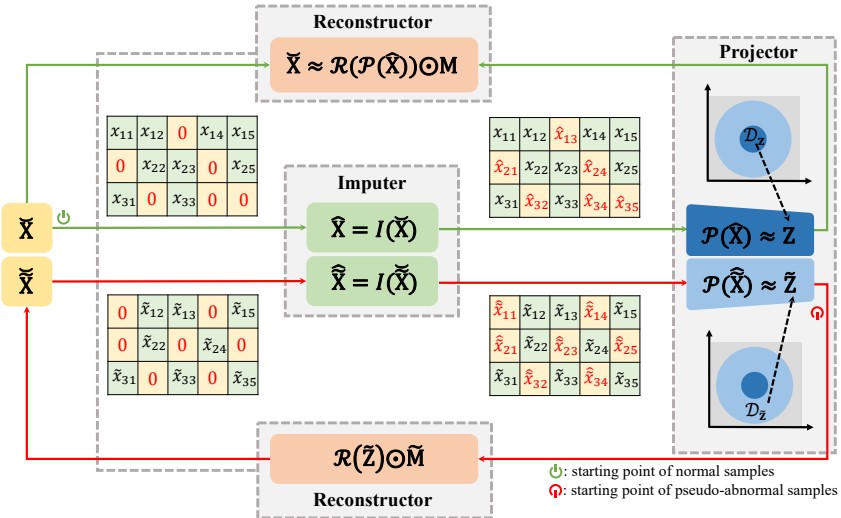

Figure 3: The architecture of ImAD, which has three modules including Imputer, Projector, and Reconstructor. Note that the two Reconstructors in the flow path share parameters. The green arrows and red arrows show the flow path of normal samples and pseudo-abnormal samples, respectively.

### 3.4 Specific Implementation

We use three neural networks $h_\psi$, $f_\theta$ and $g_\phi$ with parameters $\psi, \theta, \phi$ to model $\mathcal{I}, \mathcal{P}$ and $\mathcal{R}$ respectively. For $\mathcal{E}$, we consider two different cases. If the samples are pair-wise, we directly use the square loss, which is simple and efficient. Thus, in $\mathcal{L}^{\text{DI}}$ and $\mathcal{L}^{\text{RE}}$, we use the square loss, and the square loss for $\mathcal{L}^{\text{RE}}$ is masked by $\mathbf{M}$. When the samples are not pair-wise, we take advantage of the Sinkhorn divergence (Cuturi, 2013) derived from the optimal transport theory. The Sinkhorn divergence between two distributions $\mathcal{D}_\mathcal{U}$ and $\mathcal{D}_\mathcal{V}$ supported by their finite samples $\mathcal{U} = \{\mathbf{u}_1, \mathbf{u}_2, \cdots, \mathbf{u}_{n_u}\} \sim \mathcal{D}_u$ and $\mathcal{V} = \{\mathbf{v}_1, \mathbf{v}_2, \cdots, \mathbf{v}_{n_v}\} \sim \mathcal{D}_v$ is defined as

$$\text{Sinkhorn}(\mathcal{U},\mathcal{V}) := \min_{\mathbf{P}} \; \langle \mathbf{P}, \mathbf{C} \rangle_F + \eta \sum_{i,j} \log(P_{ij}), \;\; \text{s.t.} \; \mathbf{P}\mathbf{1} = \mathbf{a}, \mathbf{P}^T\mathbf{1} = \mathbf{b}, \mathbf{P} \geq 0 \tag{10}$$

where $\mathbf{P} \in \mathbb{R}^{n_u \times n_v}$ is the transport plan and $\mathbf{C} \in \mathbb{R}^{n_u \times n_v}$ is the metric cost matrix. The two probability vectors $\mathbf{a}$ and $\mathbf{b}$ satisfy $\mathbf{a}^T\mathbf{1} = 1, \mathbf{b}^T\mathbf{1} = 1$, and $\eta$ is a trade-off between Wasserstein distance and entropy regularization. When $\eta = 0$, Sinkhorn distance is the Wasserstein distance.

By applying $h_\psi, f_\theta, g_\phi$, square loss, ad Sinkhorn divergence to (9), we obtain the following problem:

$$\underset{\psi,\theta,\phi}{\text{minimize}} \quad \underbrace{\text{Sinkhorn}(f_\theta(h_\psi(\check{\mathbf{X}})), \mathbf{Z}) + \alpha\|\tilde{\mathbf{Z}} - f_\theta(h_\psi(g_\phi(\tilde{\mathbf{Z}}) \odot \tilde{\mathbf{M}}))\|_F^2}_{\mathcal{L}^{(\text{AD})}}$$

$$+ \underbrace{\beta\|([\check{\mathbf{X}}; \tilde{\check{\mathbf{X}}}] - h_\psi([\check{\mathbf{X}}; \tilde{\check{\mathbf{X}}}])) \odot [\mathbf{M}; \tilde{\mathbf{M}}]\|_F^2}_{\mathcal{L}^{(\text{DI})}} + \underbrace{\lambda\|(\check{\mathbf{X}} - g_\phi(f_\theta(h_\psi(\check{\mathbf{X}})))) \odot \mathbf{M}\|_F^2}_{\mathcal{L}^{(\text{RE})}} \quad (11)$$

Solving the problem (11), we get well trained imputer $h_{\psi^*}$ and projector $f_{\theta^*}$. For a new sample $\check{\mathbf{x}}_{\text{new}}$ containing missing values, we define anomaly score $s(\cdot)$ by

$$s(\check{\mathbf{x}}_{\text{new}}) = \|f_{\theta^*}(h_{\psi^*}(\check{\mathbf{x}}_{\text{new}}))\|, \quad (12)$$

which is the distance to the origin in the latent space. If $s(\check{\mathbf{x}}_{\text{new}}) > r_1$, $\check{\mathbf{x}}_{\text{new}}$ is detected as abnormal. Otherwise, $\check{\mathbf{x}}_{\text{new}}$ treated as a normal sample.

## 3.5 THEORETICAL ANALYSIS FOR LATENT SPACE SAMPLING

The theoretical analysis for the sampling strategy from the truncated Gaussians in the latent space is presented in Appendix A.

## 3.6 THEORETICAL ANALYSIS FOR GENERATION AND DETECTION ABILITY

Without loss of generality, we suppose $f_\theta$, $g_\phi$, and $h_\psi$ have $L$ layers, where $\theta = \{\mathbf{W}_1^f, \mathbf{W}_2^f, \ldots, \mathbf{W}_L^f\}$, $g = \{\mathbf{W}_1^g, \mathbf{W}_2^g, \ldots, \mathbf{W}_L^g\}$, and $\psi = \{\mathbf{W}_1^h, \mathbf{W}_2^h, \ldots, \mathbf{W}_L^h\}$. Denote the spectral norm of a matrix as $\|\cdot\|_2$. We have the following theorem (proved in Appendix B).

**Theorem 3.1.** *Suppose the activation functions in $f_\psi$, $g_\phi$, and $h_\psi$ are $\rho$-Lipschitz and $\|\mathbf{W}_l^f\|_2 \leq \alpha_f$, $\|\mathbf{W}_l^g\|_2 \leq \alpha_g$, $\|\mathbf{W}_l^h\|_2 \leq \alpha_h$, $l = 1, 2, \ldots, L$. Then:*
*(a) $\|g_\psi(\mathbf{z}) - g_\psi(\tilde{\mathbf{z}})\| \leq \rho^L \alpha_g^L \|\mathbf{z} - \tilde{\mathbf{z}}\|$ holds for any $\mathbf{z}$ and $\tilde{\mathbf{z}}$;*
*(b) $\|f_\theta(h_\psi(\check{\mathbf{x}})) - f_\theta(h_\psi(\tilde{\check{\mathbf{x}}}))\| \leq \rho^{2L} \alpha_f^L \alpha_h^L \|\check{\mathbf{x}} - \tilde{\check{\mathbf{x}}}\|$ holds for any $\check{\mathbf{x}}$ and $\tilde{\check{\mathbf{x}}}$.*

Theorem 3.1(a) indicates that in the latent space $\mathcal{Z}$, if an abnormal sample $\tilde{\mathbf{z}} \sim \mathcal{D}_{\tilde{\mathbf{z}}}$ is close to a normal sample $\mathbf{z} \sim \mathcal{D}_{\mathbf{z}}$, in the original data space, the corresponding abnormal sample $\tilde{\mathbf{x}}$ is still close to the normal sample $\mathbf{x}$ provided that $\alpha_g$ is not too large. This means the generated pseudo-abnormal samples are practical and useful. For Theorem 3.1(b), let's consider an incomplete abnormal sample $\check{\mathbf{x}}$ and assume that its closest incomplete pseudo-abnormal sample generated by the $\tilde{\mathbf{z}}$ on the outer hypersphere (shown in Figure 2) is $\tilde{\check{\mathbf{x}}}^*$, where $\|\check{\mathbf{x}} - \tilde{\check{\mathbf{x}}}^*\| = \beta$. Then in the latent space, we have $\|\tilde{\mathbf{z}} - \tilde{\mathbf{z}}^*\| \leq \rho^{2L} \alpha_f^L \alpha_h^L \beta$. Let the radii of the inner and outer hyperspheres be $r_1$ and $r_2$ respectively. Now we can conclude that if $r_2 - r_1 > \rho^{2L} \alpha_f^L \alpha_h^L \beta$, $\tilde{\mathbf{z}}$ is outside the decision region given by the inner hypersphere and hence $\tilde{\check{\mathbf{x}}}$ is successfully detected as an abnormal sample.

## 4 EXPERIMENTS

We compare our method with "impute-then-detect" methods on seven publicly available tabular datasets. In all experiments, only incomplete normal data are used in the training stage, but there are both incomplete normal and abnormal data in the testing stage. Due to space limitation, we report the main experimental results in this section. In Appendix C, we explore the gain of introduced pseudo-abnormal samples for detection performance. Furthermore, we also analyze the influences of the constrained radii $r_1, r_2$ for detection performance and related results are provided in Appendix D. We reported the remaining results under different experimental settings in Appendix G.

## 4.1 DATASETS AND BASELINES

The statistic information of all datasets used in our experiments is provided in Table 1 and detailed description of all datasets is put in the Appendix F.

Table 1: Statistics of the seven datasets (Samp. means Samples).

| Name | Field | Dimension | Instances | Normal Samp. | Abnormal Samp. |
|---|---|---|---|---|---|
| Adult | income census | 14 | 30,162 | 22,658 | 7,508 |
| Botnet | cybersecurity | 115 | 40,607 | 13,113 | 27,494 |
| KDD | cybersecurity | 121 | 494,021 | 396,743 | 97,278 |
| Arrhythmia | medical diagnosis | 274 | 452 | 320 | 132 |
| Speech | speech recognition | 400 | 3,686 | 3,625 | 61 |
| Segerstolpe | cell analysis | 1,000 | 702 | 329 | 372 |
| Usoskin | cell analysis | 25,334 | 610 | 232 | 378 |

In terms of "impute-then-detect", we consider both conventional machine learning based methods and deep learning based methods. For imputation, we use MissForest (Stekhoven & Bühlmann, 2012) and GAIN (Yoon et al., 2018). For anomaly detection, we use Isolation Forest (Liu et al., 2008), Deep SVDD (Ruff et al., 2018), and NeutraL AD (Qiu et al., 2021). Concatenating imputation methods and anomaly detection methods pairwise, we get six two-stage baselines.

## 4.2 Implementation details

On all experimental datasets, we use MLPs for all three modules of ImAD, including the imputation module, projection module, and reconstruction module. We use Adam (Kingma & Ba, 2015) as the optimizer and set coefficient $\eta$ of entropy regularization term in Sinkhorn distance to 0.1 in all experiments. Other experimental hyper-parameters are provided in Appendix F. Sensitivity analysis of hyper-parameters is provided in Appendix E. The detailed description of missing mechanisms, including MCAR, MAR, and MNAR, is provided in Appendix F. In all experiments, we set missing rate $r = \{0.2, 0.5\}$. For Adult and KDD dataset, we both consider two different test set splitting strategies and results under skewed splitting are reported in Appendix G.

We use the AUROC (Area Under the Receiver Operating Characteristic curve) and AUPRC (Area Under the Precision-Recall curve) to evaluate the detection performance. ALL experiments were conducted on 20 Cores Intel(R) Xeon(R) Gold 6248 CPU with one NVIDIA Tesla V100 GPU, CUDA 12.0. We repeat the experiment of each setting five times and report the average performance.

## 4.3 Experimental results under MCAR

The results of anomaly detection with missing data under the setting of MCAR are shown in Table 2, Table 3, Table 4 and Table 5. We have the following observations:

- On all datasets, the detection performance in terms of APROC and AUPRC of two-stage methods do not decrease correspondingly with the increasing of missing rate $r$ in some cases, which reflects the negative influences of imputation bias for two-stage pipeline methods. In other words, a smaller missing rate means a simpler imputation task and more serious imputation bias, which makes the detection performance of the two-stage methods suffer from significant degradation when the missing rate $r$ is small.

- For the two-stage methods, "MissForest" outperformed "GAIN" in most cases, which indicates that a state-of-the-art imputation method may not bring positive gain for unsupervised anomaly detection problem in the presence of missing values instead the outstanding recovery ability because the identical distribution assumption does not hold.

- On all datasets, ImAD has the best detection performance in almost all cases. It is worth noting that the detection performance of ImAD decreases corresponding with the increasing missing rate in all cases, which indicates that ImAD does not suffer the significant impacts from imputation bias and the generated pseudo-abnormal samples are effective for alleviating the learning bias.

In Figure 4, we visualize the projection results of Botnet dataset in 2-D space. It can be observed that the majority of normal training and testing samples are mapped into the target distribution while most abnormal samples fall outside of the decision boundary. This demonstrates that our method is practical and Theorem 3.1 is effective for real-world scenarios.

Table 2: AUROC and AUPRC (%, mean and std) on KDD and Adult datasets with MCAR under balanced data splitting. The best result in each case is marked in **bold**.

| DI Methods | AD Methods | KDD | | | | Adult | | | |
|---|---|---|---|---|---|---|---|---|---|
| | | AUROC(%) | | AUPRC(%) | | AUROC(%) | | AUPRC(%) | |
| | | $r = 0.2$ | $r = 0.5$ | $r = 0.2$ | $r = 0.5$ | $r = 0.2$ | $r = 0.5$ | $r = 0.2$ | $r = 0.5$ |
| MissForest | I-Forest (Liu et al., 2008) | 94.90 (1.95) | **93.37** (2.15) | 93.24 (2.38) | 93.21 (1.92) | 60.06 (1.69) | 60.73 (0.69) | 57.12 (2.16) | 56.80 (1.27) |
| | Deep SVDD (Ruff et al., 2018) | 93.58 (2.46) | 91.84 (5.91) | 85.77 (2.95) | 88.79 (1.29) | 62.33 (4.86) | 61.21 (2.24) | 55.31 (2.91) | 55.45 (1.72) |
| | NeutraL AD (Qiu et al., 2021) | 94.00 (1.72) | 92.68 (2.44) | 93.87 (1.57) | **94.88** (2.86) | 58.79 (1.88) | 55.12 (3.41) | 50.07 (6.50) | 52.27 (3.61) |
| GAIN | I-Forest (Liu et al., 2008) | 82.78 (3.80) | 79.94 (0.39) | 90.33 (1.58) | 89.52 (1.07) | 59.53 (0.91) | 61.18 (1.61) | 57.05 (1.02) | 56.87 (1.09) |
| | Deep SVDD (Ruff et al., 2018) | 88.68 (4.87) | 88.44 (5.54) | 88.36 (3.42) | 85.45 (5.67) | 58.65 (3.44) | 65.44 (2.40) | 57.61 (4.24) | 59.55 (2.34)) |
| | NeutraL AD (Qiu et al., 2021) | 90.48 (3.24) | 84.10 (0.91) | 84.61 (1.30) | 84.08 (1.71) | 55.04 (1.81) | 56.44 (2.13) | 53.00 (6.80) | 59.06 (3.97) |
| ImAD (Ours) | | **97.01** (0.33) | 90.78 (1.35) | **95.96** (0.18) | 91.58 (0.32) | **76.51** (2.12) | **71.19** (1.63) | **73.42** (2.08) | **71.50** (2.02) |

Table 3: AUROC and AUPRC (%, mean and std) on Arrhythmia and Speech datasets with MCAR under balanced data splitting.

| DI Methods | AD Methods | Arrhythmia | | | | Speech | | | |
|---|---|---|---|---|---|---|---|---|---|
| | | AUROC(%) | | AUPRC(%) | | AUROC(%) | | AUPRC(%) | |
| | | $r = 0.2$ | $r = 0.5$ | $r = 0.2$ | $r = 0.5$ | $r = 0.2$ | $r = 0.5$ | $r = 0.2$ | $r = 0.5$ |
| MissForest | I-Forest (Liu et al., 2008) | 80.72 (0.62) | 81.54 (0.95) | 77.91 (1.85) | 77.95 (0.97) | 28.58 (2.95) | 29.09 (1.14) | 36.83 (1.06) | 37.29 (0.76) |
| | Deep SVDD (Ruff et al., 2018) | 72.63 (0.99) | 75.80 (4.07) | 70.94 (0.75) | 77.39 (4.55) | 60.37 (0.87) | 40.14 (4.30) | 58.93 (1.35) | 42.08 (2.16) |
| | NeutraL AD (Qiu et al., 2021) | 47.38 (4.81) | 44.30 (2.11) | 50.87 (3.53) | 50.12 (2.52) | 56.51 (4.87) | 54.11 (3.77) | 55.44 (4.36) | 52.26 (3.97) |
| GAIN | I-Forest (Liu et al., 2008) | 77.19 (0.81) | 76.29 (1.35) | 76.40 (1.86) | 76.29 (1.35) | 29.33 (0.59) | 29.23 (2.13) | 39.92 (0.21) | 40.04 (0.63) |
| | Deep SVDD (Ruff et al., 2018) | 57.14 (5.41) | 48.86 (2.35) | 59.35 (2.58) | 54.03 (2.45) | 54.95 (1.79) | 46.54 (2.10) | 54.38 (0.96) | 47.54 (1.75) |
| | NeutraL AD (Qiu et al., 2021) | 37.96 (5.09) | 33.98 (4.12) | 42.57 (2.56) | 42.35 (1.96) | 56.80 (4.89) | 57.24 (5.51) | 54.76 (4.58) | 55.05 (5.58) |
| ImAD (Ours) | | **82.24** (1.76) | **81.76** (1.19) | **83.74** (1.85) | **83.37** (1.36) | **61.94** (2.77) | **58.66** (1.40) | **60.43** (3.33) | **58.13** (1.48) |

Table 4: AUROC and AUPRC (%, mean and std) on Segerstolpe and Usoskin datasets with MCAR under balanced data splitting.

| DI Methods | AD Methods | Arrhythmia | | | | Speech | | | |
|---|---|---|---|---|---|---|---|---|---|
| | | AUROC(%) | | AUPRC(%) | | AUROC(%) | | AUPRC(%) | |
| | | $r = 0.2$ | $r = 0.5$ | $r = 0.2$ | $r = 0.5$ | $r = 0.2$ | $r = 0.5$ | $r = 0.2$ | $r = 0.5$ |
| MissForest | I-Forest (Liu et al., 2008) | 94.91 (1.35) | 96.68 (0.79) | 95.94 (1.23) | **97.56** (0.59) | 45.19 (4.56) | 49.64 (7.43) | 46.97 (3.04) | 49.74 (5.64) |
| | Deep SVDD (Ruff et al., 2018) | 96.20 (2.66) | 89.24 (1.44) | 97.53 (1.40) | 90.65 (0.57) | 37.47 (3.83) | 43.61 (7.49) | 50.55 (2.02) | 55.05 (4.81) |
| | NeutraL AD (Qiu et al., 2021) | 97.89 (1.45) | 89.38 (2.80) | 97.71 (1.76) | 84.61 (3.78) | 57.43 (4.59) | 53.74 (2.27) | 63.65 (2.40) | 61.05 (4.16) |
| GAIN | I-Forest (Liu et al., 2008) | 94.25 (0.90) | 92.07 (1.82) | 96.14 (0.75) | 93.94 (1.62) | 40.96 (2.02) | 37.11 (2.12) | 46.29 (1.76) | 42.86 (1.22) |
| | Deep SVDD (Ruff et al., 2018) | 92.46 (4.25) | 94.32 (1.93) | 92.25 (2.40) | 92.88 (1.26) | 49.99 (5.69) | 65.48 (2.94) | 54.85 (1.61) | 64.54 (0.74) |
| | NeutraL AD (Qiu et al., 2021) | 97.52 (0.37) | 90.10 (0.90) | 97.52 (1.02) | 90.10 (0.82) | 56.18 (2.62) | 64.80 (1.85) | 64.85 (2.68) | 73.33 (1.31) |
| ImAD (Ours) | | **99.14** (0.88) | **96.86** (0.67) | **98.98** (1.18) | 96.85 (0.54) | **84.95** (1.29) | **79.23** (2.49) | **85.48** (2.34) | **80.06** (3.40) |

## 4.4 EXPERIMENTAL RESULTS UNDER MAR AND MNAR

Table 5: AUROC, AUPRC (%, mean and std) on Botnet dataset with MCAR.

| DI Methods | AD Methods | AUROC | | AUPRC | |
|---|---|---|---|---|---|
| | | $r = 0.2$ | $r = 0.5$ | $r = 0.2$ | $r = 0.5$ |
| MissForest | I-Forest (Liu et al., 2008) | 95.72 (0.96) | 93.86 (0.70) | 97.25 (0.69) | 95.68 (0.52) |
| | Deep SVDD (Ruff et al., 2018) | 96.72 (0.87) | 97.51 (0.94) | 96.60 (0.80) | 97.62 (0.89) |
| | NeutraL AD (Qiu et al., 2021) | 99.04 (0.26) | 97.27 (0.59) | 98.92 (0.24) | 97.68 (0.53) |
| GAIN | I-Forest (Liu et al., 2008) | 96.16 (0.24) | 94.01 (0.73) | 97.61 (0.21) | 96.18 (0.44) |
| | Deep SVDD (Ruff et al., 2018) | 98.68 (0.11) | 98.02 (0.41) | 98.35 (0.14) | 97.59 (0.46) |
| | NeutraL AD (Qiu et al., 2021) | 97.42 (0.33) | **99.56** (0.27) | 96.89 (0.36) | 99.41 (0.35) |
| ImAD (Ours) | | **99.71** (0.22) | 99.53 (0.25) | **99.68** (0.24) | **99.58** (0.20) |

Table 6 displays the results on Adult dataset with missing mechanisms MAR and MNAR. We did not use GAIN (Yoon et al., 2018) in theses two cases because GAIN (Yoon et al., 2018) is proposed under the MCAR assumption. Instead, we use MissOT (Muzellec et al., 2020) in the two-stage methods. Due to space limitation, we only report the average AUROC(%) without standard deviation for all baselines in Table 6 and the results with standard deviation are provided in Appendix H. As shown by Table 6, ImAD outperforms all two-stage methods significantly in both MAR and MNAR settings. Consistent with MCAR, on both MAR and MNAR, the detection performance of the two-stage methods do not decrease correspondingly with the increasing of missing rate $r$ in some cases.

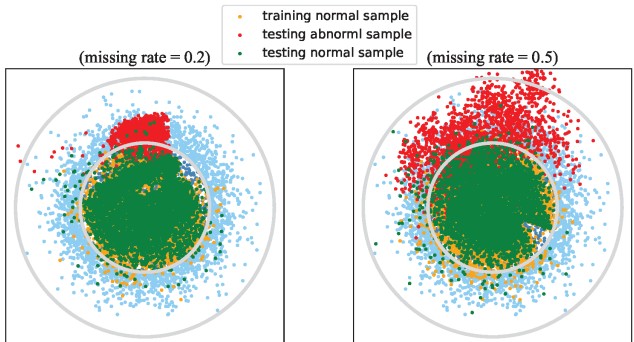

Figure 4: Visualization of projection results in 2-dimensional space on Botnet dataset.

Table 6: Average AUROC(%) on Adult dataset with MAR and MNAR.

| DI Methods | AD Methods | MAR | | | | MNAR | | | |
|---|---|---|---|---|---|---|---|---|---|
| | | Balanced Split | | Skewed Split | | Balanced Split | | Skewed Split | |
| | | $r = 0.2$ | $r = 0.5$ | $r = 0.2$ | $r = 0.5$ | $r = 0.2$ | $r = 0.5$ | $r = 0.2$ | $r = 0.5$ |
| MissForest | I-Forest (Liu et al., 2008) | 60.54 | 61.94 | 63.07 | 63.17 | 60.53 | 60.24 | 61.35 | 61.35 |
| | Deep SVDD (Ruff et al., 2018) | 61.53 | 56.22 | 57.55 | 57.00 | 54.90 | 57.54 | 56.27 | 58.50 |
| | NeutraL AD (Qiu et al., 2021) | 52.29 | 51.96 | 54.37 | 53.32 | 53.07 | 50.82 | 52.97 | 52.63 |
| MissOT$_{(MLP)}$ | I-Forest (Liu et al., 2008) | 45.63 | 41.94 | 45.32 | 42.44 | 44.78 | 38.62 | 44.53 | 38.78 |
| | Deep SVDD (Ruff et al., 2018) | 51.68 | 39.59 | 44.49 | 50.04 | 45.77 | 50.29 | 51.47 | 49.12 |
| | NeutraL AD (Qiu et al., 2021) | 52.54 | 47.24 | 52.32 | 44.75 | 49.87 | 49.38 | 49.85 | 49.62 |
| ImAD (Ours) | | **77.43** (3.42) | **74.61** (2.18) | **80.61** (2.13) | **73.68** (2.10) | **73.73** (3.57) | **72.35** (1.53) | **76.10** (2.04) | **75.58** (2.44) |

## 5 CONCLUSION

This paper proposed ImAD, the first end-to-end unsupervised anomaly detection method in the presence of missing values. ImAD integrates data imputation and anomaly detection into a unified optimization objective and introduces pseudo-abnormal samples to alleviate the imputation bias. We conducted experiments on seven real-world datasets and considered three different missing mechanisms, including MCAR, MAR and MNAR. The results indicated that ImAD effectively alleviates the imputation bias and achieves better detection performance than the two-stage "impute-then-detect" methods in almost all cases.

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

## A   SAMPLING ANALYSIS

In this section, for target distribution $\mathcal{D}_{\mathbf{z}}, \mathcal{D}_{\tilde{\mathbf{z}}} \sim \mathcal{N}(\mathbf{0}, \sigma^2 \mathbf{I}_d)$, we give the lower bound of the constrained sampling radius $r$ when given a sampling probability $p$.

For target distribution $\mathcal{D}_{\mathbf{z}}$, we expect that it is compact and can be easily sampled, in which the compactness is to ensure a clear and reliable decision boundary between normal and abnormal data. Therefore, we select truncated Gaussian from $\mathcal{N}(\mathbf{0}, \sigma^2 \mathbf{I}_d)$ as target distribution $\mathcal{D}_{\mathbf{z}}$ and bound $\mathcal{D}_{\mathbf{z}}$ in a $d$-dimensional radius-$r$ hyperball centering at origin. For radius $r$, we have the following proposition.

**Proposition A.1.** *Let $F_d$ denote the cumulative distribution function (CDF) of the chi-square distribution $\chi^2(d)$. For a given probability $0 < p < 1$, when $r \geq \sigma\sqrt{F_d^{-1}(p)}$, the sampling probability in $\mathcal{D}_{\mathbf{z}}$ satisfies $P(\|\mathbf{x}\|^2 < r^2) \geq p$ where $\mathbf{x} = [x_1, x_2, \cdots, x_d]$ and $x_1, \ldots, x_d \overset{i.i.d.}{\sim} \mathcal{N}(0, \sigma^2)$.*

*Proof.*

$$\text{We have } x_1, \ldots, x_d \overset{\text{i.i.d.}}{\sim} \mathcal{N}(0, \sigma^2) \implies \frac{x_1}{\sigma}, \ldots, \frac{x_d}{\sigma} \overset{\text{i.i.d.}}{\sim} \mathcal{N}(0, 1) \implies \frac{\sum_{i=1}^{d} x_i^2}{\sigma^2} \sim \chi^2(d).$$

$$\text{Let } Y = \frac{\sum_{i=1}^{d} x_i^2}{\sigma^2}, \text{ we get } P\left(Y < F_d^{-1}(p)\right) = p$$

$$\implies P\left(\frac{\sum_{i=1}^{d} x_i^2}{\sigma^2} < F_d^{-1}(p)\right) = p \tag{13}$$

$$\implies P\left(\sum_{i=1}^{d} x_i^2 < \sigma^2 \cdot F_d^{-1}(p)\right) = p$$

$$\implies P\left(\|\mathbf{x}\|^2 < \left(\sigma\sqrt{F_d^{-1}(p)}\right)^2\right) = p.$$

$$\text{Therefore, } r \geq \sigma\sqrt{F_d^{-1}(p)} \implies P\left(\|\mathbf{x}\|^2 < r^2\right) \geq p.$$

$\square$

In keeping with target distribution $\mathcal{D}_{\mathbf{z}}$, we select truncated Gaussian from $\mathcal{N}(\mathbf{0}, \sigma^2 \mathbf{I}_d)$ as target distribution $\mathcal{D}_{\tilde{\mathbf{z}}}$ and bound $\mathcal{D}_{\tilde{\mathbf{z}}}$ between two $d$-dimensional hyperspheres with radii $r_1, r_2$ respectively, cetering at origin, where $r_2 > r_1$. For radius $r_1, r_2$, we have the following proposition.

**Proposition A.2.** *Let $F_d$ denote the cumulative distribution function (CDF) of the chi-square distribution $\chi^2(d)$. For a given probability $0 < p < 1$, when $r_1 \leq \sigma\sqrt{F_d^{-1}(p_1)}, r_2 \geq \sigma\sqrt{F_d^{-1}(p_2)}$ and satisfies $p = p_2 - p_1$, the sampling probability in $\mathcal{D}_{\tilde{\mathbf{z}}}$ satisfies $P(r_1^2 < \|\mathbf{x}\|^2 < r_2^2) \geq p$ where $\mathbf{x} = [x_1, x_2, \cdots, x_d]$ and $x_1, \ldots, x_d \overset{i.i.d.}{\sim} \mathcal{N}(0, \sigma^2)$.*

*Proof.* According the proof for Proposition A.1, we have

$$r \geq \sigma\sqrt{F_d^{-1}(p)} \Longrightarrow P(\|\mathbf{x}\|^2 < r^2) \geq p.$$

$$\text{Therefore, } r_1 \leq \sigma\sqrt{F_d^{-1}(p_1)} \Longrightarrow P(\|\mathbf{x}\|^2 < r_1^2) \leq p_1,$$

$$\text{and } r_2 \geq \sigma\sqrt{F_d^{-1}(p_2)} \Longrightarrow P(\|\mathbf{x}\|^2 < r_2^2) \geq p_2.$$

$$\text{Therefore, } P\left(\|\mathbf{x}\|^2 < r_2^2\right) - P(\|\mathbf{x}\|^2 < r_1^2)$$

$$= P\left(r_1^2 < \|\mathbf{x}\|^2 < r_2^2\right) \geq p_2 - p_1 = p \tag{14}$$

$\square$

As shown in Figure 2, we set radius $r_1$ of $\mathcal{D}_{\tilde{\mathbf{z}}}$ equals to radius $r$ of $\mathcal{D}_{\mathbf{z}}$. Also, we maintains such settings $r_1 = r$ in our experiments to make the introduced pseudo-abnormal samples are not far from the normal data.

## B  PROOF FOR THEOREM 3.1

*Proof.* Recall that $g_\psi$ was defined as

$$g_\psi(\mathbf{z}) = \sigma_L(\mathbf{W}_L^g(\cdots\sigma_2(\mathbf{W}_2^g(\sigma_1(\mathbf{W}_1^g\mathbf{z})))\cdots)). \tag{15}$$

Then for any $\mathbf{z}, \tilde{\mathbf{z}} \in \mathbb{R}^d$, we have

$$\begin{aligned}
&\|g_\psi(\mathbf{z}) - g_\psi(\tilde{\mathbf{z}})\| \\
=&\|\sigma_L(\mathbf{W}_L^g(\cdots\sigma_2(\mathbf{W}_2^g(\sigma_1(\mathbf{W}_1^g\mathbf{z})))\cdots)) - \sigma_L(\mathbf{W}_L^g(\cdots\sigma_2(\mathbf{W}_2^g(\sigma_1(\mathbf{W}_1^g\tilde{\mathbf{z}})))\cdots))\| \\
\leq&\rho\|\mathbf{W}_L^g(\cdots\sigma_2(\mathbf{W}_2^g(\sigma_1(\mathbf{W}_1^g\mathbf{z})))\cdots) - \mathbf{W}_L^g(\cdots\sigma_2(\mathbf{W}_2^g(\sigma_1(\mathbf{W}_1^g\tilde{\mathbf{z}})))\cdots)\| \\
\leq&\rho\|\mathbf{W}_L^g\|_2\|\sigma_{L-1}(\cdots\sigma_2(\mathbf{W}_2^g(\sigma_1(\mathbf{W}_1^g\mathbf{z})))\cdots) - \sigma_{L-1}(\cdots\sigma_2(\mathbf{W}_2^g(\sigma_1(\mathbf{W}_1^g\tilde{\mathbf{z}})))\cdots)\| \\
\leq&\rho^L\left(\prod_{l=1}^{L}\|\mathbf{W}_l^g\|_2\right)\|\mathbf{z} - \tilde{\mathbf{z}}\| \\
\leq&\rho^L\alpha_g^L\|\mathbf{z} - \tilde{\mathbf{z}}\|.
\end{aligned} \tag{16}$$

This finished the proof for part (a) of the theorem. The proof for part (b) is similar and omitted here for simplicity. $\square$

## C  GAIN OF DETECTION PERFORMANCE FROM PSEUDO-ABNORMAL SAMPLES

In this section, we explore the influences of introduced pseudo-abnormal samples for detection performance. Both on Adult and KDD datasets, we remove the pseudo-abnormal samples in training process and only use incomplete normal data to training ImAD. The experimental results are showed in Table 7. Observing the results in Table 7, the detection performance of ImAD is improved on all the cases when introducing pseudo-abnormal samples into training process, which indicates that the pseudo-abnormal samples bring positive gains for ImAD.

Table 7: Gain of detection performance of ImAD from pseudo-abnormal samples under MCAR.

| Datasets | Settings | Balanced Split | | Skewed Split | |
|---|---|---|---|---|---|
| | | $r$=0.2 | $r$=0.5 | $r$=0.2 | $r$=0.5 |
| Adult | ImAD w/o pseudo-abnormal samples | 72.04 | 69.07 | 70.03 | 69.73 |
| | ImAD | 76.51 | 71.19 | 74.65 | 71.02 |
| KDD | ImAD w/o pseudo-abnormal samples | 95.89 | 88.63 | 94.90 | 92.77 |
| | ImAD | 97.01 | 90.78 | 98.41 | 93.62 |

# D  THE INFLUENCE OF CONSTRAINED RADII $r_1, r_2$ FOR DETECTION PERFORMANCE

In this section, we explore the influences of constrained radii $r_1, r_2$ for detection performance. We change the latent dimension $d = \{16, 32, 64, 128, 256\}$ and conduct related experiments on Arrhythmia dataset. More experimental details and results are provided in Table 8 and Figure 5.

As showed in Table 8, we change the dimension $d$ of latent space and then get $r = \sigma\sqrt{F_d^{-1}(p)}$ (See Proposition A.1) and set target distribution $\mathcal{D}_{\mathbf{z}} \sim \mathcal{N}(\mathbf{0}, 0.5^2 \cdot \mathbf{I}_d), \mathcal{D}_{\bar{\mathbf{z}}} \sim \mathcal{N}(\mathbf{0}, \mathbf{I}_d)$ and set $p = 0.9$.

Table 8: The constrained radii $r_1, r_2$ under with different latent dimensions.

| Radius | Latent Dimension ($d$) | | | | |
|---|---|---|---|---|---|
| | 16 | 32 | 64 | 128 | 256 |
| $r_1 = 0.5\sqrt{F_d^{-1}(0.9)}$ | 2.42 | 3.26 | 4.44 | 6.10 | 8.45 |
| $r_2 = \sqrt{F_d^{-1}(0.9)}$ | 4.85 | 6.52 | 8.88 | 12.20 | 16.90 |

Figure 5 shows the average AUROC(%) and AUPRC(%) on Arrhythmia dataset under balanced data splitting when changing latent dimension $d$. It can be observed that our method is not quite sensitive to the changes of radii $r_1, r_2$ and the performance declines with the decrease of the latent dimension, which is reasonable since there will be more information loss when the latent dimension becomes smaller.

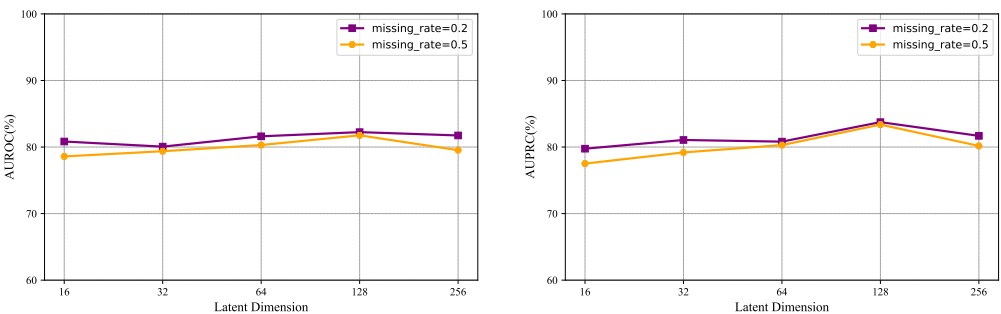

Figure 5: The performance fluctuation on Arrhythmia with different latent dimension.

# E  ABLATION STUDY AND HYPERPARAMETER ANALYSIS

For hyper-parameters $\alpha, \beta, \lambda$ used in our experiments, we vary them in a large range to analyze the sensitivity of ImAD and set missing rate $r = 0.2$ (MCAR) in all experiments. For hyper-parameter $\beta$, it cannot be set to 0 because the imputation module is an indispensable part in the presence of missing values. The average results are shown in Figure 6, where (a), (b), (c) illustrate the fluctuation of detection performance on balanced splitting, and (d), (e), (f) illustrate the fluctuation of detection performance on skewed splitting.

# F  DETAILED EXPERIMENTAL IMPLEMENTATIONS

**Dataset Description.**

- **Adult**[1] (Becker & Kohavi, 1996) is from the 1994 Census Income database with 14 variables including both categorical and continuous variables. The samples of income $\leq 50K$ are regarded as normal data, and the samples of income $> 50K$ are regarded as abnormal data. Data preparation follows the previous work (Han et al., 2023).

---

[1]https://archive.ics.uci.edu/dataset/2/adult

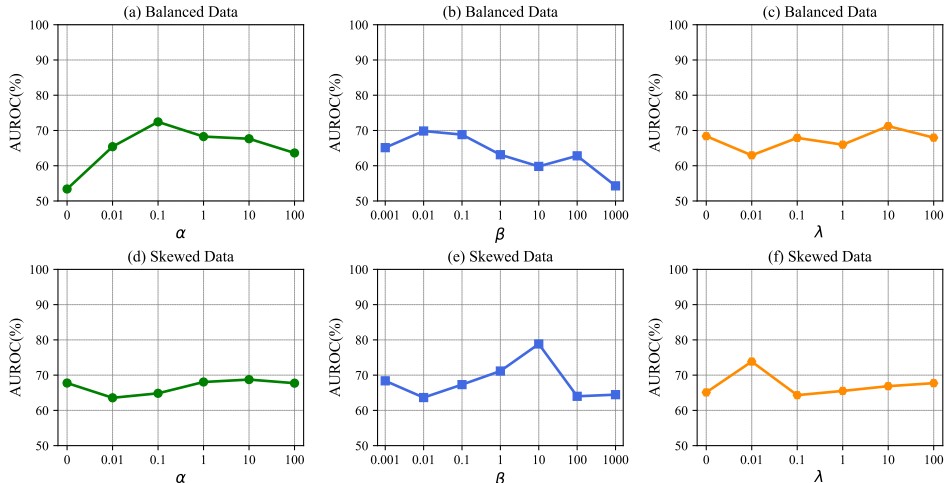

Figure 6: Sensitivity analysis of hyperparameters $\alpha, \beta, \lambda$ on Adult dataset.

- **KDD**[2](Lichman, 2013) is the KDDCUP99 10 percent dataset from the UCI repository and contains 121 variables including both categorical and continuous variables. The attack samples are regarded as normal data, and the non-attack samples are regarded as abnormal data.

- **Arrhythmia**[3] (Rayana, 2016) is an ECG dataset. It was used to identify arrhythmic samples in five classes and contains 452 instances with 274 attributes.

- **Speech**[4] (Rayana, 2016) consists of 3686 segments of English speech spoken with different accents and is represented by 400-dimensional so called i-vectors which are widely used state-of-the-art features for speaker and language recognition.

- **Segerstolpe** (Segerstolpe et al., 2016) is an scRNA-seq dataset of human pancreas islets which includes six cell types: "alpha", "beta", "delta", "ductal", "endothelial" and "gamma". In our experiments, "alpha" is regarded as normal data and "beta" is regarded as abnormal data.

- **Usoskin** (Usoskin et al., 2015) is a dataset employed for the analysis of sensory neuron cells, specifically originating from the mouse lumbar dorsal root ganglion. The dataset encompasses four distinct cell types: non-peptidergic nociceptor cells (NP), peptidergic nociceptor cells (PEP), neurofilament-containing cells (NF), and tyrosine hydroxylase containing cells (TH). In our experimental, TH is regarded as normal data and PEP is abnormal data.

- **Botnet**[5] (Meidan & Shabtai, 2018) is a public botnet datasets for the IoT. it was gathered from 9 commercial IoT devices authentically infected by Mirai and BASHLITE. There are 7,062,606 instances in the original datasets. In our experiments, we use "Ecobee_Thermostat"subset of the original data, in which "'benign_traffic" is regarded as normal data and "gafgyt_attacks" is regarded as abnormal data. "gafgyt_attacks" has five attack types and we randomly select 1,000 samples from each type as abnormal data of test set.

**Missing Mechanism.** In this work, we evaluate detection performance of all the baselines under the three different missing mechanisms and we follow the previous work (Muzellec et al., 2020) to set missing value generation mechanism.

Detailed explanation in our implementation is provided as follows.

---

[2]https://kdd.ics.uci.edu/databases/kddcup99/

[3]http://odds.cs.stonybrook.edu/arrhythmia-dataset/

[4]https://odds.cs.stonybrook.edu/speech-dataset/

[5]https://archive.ics.uci.edu/dataset/442/detection+of+iot+botnet+attacks+n+baiot

- **MCAR**: missing completely at random if the missingness is independent of the data. In our implementation, each entry is masked according to the realization of a Bernoulli random variable with parameter $p = \{0.2, 0.5\}$.

- **MAR**: missing at random if the missingness depends only on the observed values. In MAR setting, for all experiments, a fixed subset of variables that cannot have missing values is sampled. Then, the entries from remaining variables are masked according a logistic model with random weights, which takes the non-missing variables as inputs. A bias term is fitted using line search to attain the desired proportion of missing values.

- **MNAR**: missing not at random if the missingness depends on both the observed values and the unobserved values. In MNAR setting, first, we sample a subset of variables whose values in the lower and upper p-th percentiles are masked according to a Bernoulli random variable, and the values in-between are left not missing.

**Sampling in Target Distribution.** In our experiments, we select two truncated Gaussian distribution $\mathcal{N}(\mathbf{0}, \sigma^2 \mathbf{I}_d)$ with different $\sigma$ as target distribution $\mathcal{D}_{\mathbf{z}}, \mathcal{D}_{\tilde{\mathbf{z}}}$ and set $\sigma = 0.5, \sigma = 1.0$ respectively. For target distribution $\mathcal{D}_{\mathbf{z}} \sim \mathcal{N}(\mathbf{0}, 0.5^2 \cdot \mathbf{I}_d)$, according to the Proposition A.1, we set constrained radius $r = 0.5\sqrt{F_d^{-1}(p)}$ where $d$ denotes the latent dimension and set $p = 0.9$. Similarity, for target distribution $\mathcal{D}_{\tilde{\mathbf{z}}} \sim \mathcal{N}(\mathbf{0}, \mathbf{I}_d)$, we set $r_1 = r$ and $r_2 = \sqrt{F_d^{-1}(p)}$ and set $p = 0.9$.

**All Baselines.** For data imputation method used in our experiments, GAIN [6], MissOT [7], we use official code and the hyperparameters are fine-tuned as suggested in the original paper. For MissForest, we use *missingpy* [8] that is a library for missing data imputation in Python to implement MissForest algorithm. For anomaly detection method, Deep SVDD [9], NeutraL AD [10], we use official code and the hyperparameters are fine-tuned as suggested in the original paper. For Isolation Forest, we use *scikit-learn* [11] to implement Isolation Forest algorithm.

**Hyper-parameters.** The hyperparameters used in our experiments are provided in Table 9.

Table 9: Hyperparameters settings of the proposed method on all datasets.

| Datasets | Missing rate | Latent dimension | Learning rate | $\alpha$ | $\beta$ | $\lambda$ |
|---|---|---|---|---|---|---|
| Adult | $r=0.2$ | 4 | 0.0002 | 5 | 20 | 1 |
| | $r=0.5$ | 4 | 0.0002 | 1 | 10 | 2 |
| Botnet | $r=0.2$ | 32 | 0.0001 | 1 | 1 | 1 |
| | $r=0.5$ | 32 | 0.0001 | 1 | 1 | 1 |
| KDD | $r=0.2$ | 32 | 0.0001 | 1 | 5 | 1 |
| | $r=0.5$ | 32 | 0.0001 | 1 | 5 | 1 |
| Arrhythmia | $r=0.2$ | 128 | 0.0001 | 1 | 1 | 1 |
| | $r=0.5$ | 128 | 0.0001 | 1 | 1 | 1 |
| Speech | $r=0.2$ | 128 | 0.0005 | 0.2 | 0.1 | 1 |
| | $r=0.5$ | 128 | 0.0005 | 0.2 | 0.2 | 1 |
| Segerstolpe | $r=0.2$ | 128 | 0.0001 | 1 | 1 | 1 |
| | $r=0.5$ | 128 | 0.0001 | 1 | 1 | 1 |
| Usoskin | $r=0.2$ | 128 | 0.0001 | 0.2 | 0.2 | 0.2 |
| | $r=0.5$ | 128 | 0.0001 | 0.2 | 0.2 | 0.2 |

## G  THE EXPERIMENTAL RESULTS UNDER SKEWED DATA SPLITTING.

For dataset Adult and KDD, we consider two different test set splitting strategies including balanced splitting and skewed splitting. Balanced splitting means that there are an equal number of normal

---

[6]https://github.com/jsyoon0823/GAIN

[7]https://github.com/BorisMuzellec/MissingDataOT

[8]https://pypi.org/project/missingpy/

[9]https://github.com/lukasruff/Deep-SVDD-PyTorch

[10]https://github.com/boschresearch/NeuTraL-AD

[11]https://scikit-learn.org/stable/

and abnormal samples in the test set. Skewed split means that there are more normal samples than abnormal samples in the test set which often occurs in the real-world scenarios. In our experiments, we set that there are ten times as many normal samples as abnormal samples. Table 10 reports the results on KDD and Adult datasets under skewed data splitting.

Table 10: AUROC (%, mean and std) on KDD and Adult dataset with MCAR under skewed data splitting.

| DI Methods | AD Methods | KDD | | Adult | |
|---|---|---|---|---|---|
| | | $r = 0.2$ | $r = 0.5$ | $r = 0.2$ | $r = 0.5$ |
| MissForest | I-Forest (Liu et al., 2008) | 90.44 (4.62) | 86.62 (5.30) | 62.19 (0.86) | 59.92 (0.62) |
| | Deep SVDD (Ruff et al., 2018) | 82.70 (7.61) | 88.06 (6.29) | 59.68 (4.06) | 58.56 (4.94) |
| | NeutraL AD (Qiu et al., 2021) | 91.42 (1.54) | 93.46 (4.98) | 52.97 (1.40) | 54.12 (1.90) |
| GAIN | I-Forest (Liu et al., 2008) | 79.50 (1.06) | 80.15 (1.57) | 61.60 (1.89) | 59.38 (1.90) |
| | Deep SVDD (Ruff et al., 2018) | 81.61 (3.42) | 85.84 (5.67) | 62.93 (2.67) | 65.07 (6.74) |
| | NeutraL AD (Qiu et al., 2021) | 83.33 (2.12) | 78.86 (3.60) | 56.14 (1.97) | 57.97 (2.43) |
| ImAD (Ours) | | **98.41** (0.15) | **93.62** (0.45) | **74.65** (2.37) | **71.02** (0.39) |

## H  AVERAGE AUROC WITH STANDARD DEVIATION OF BASELINES UNDER MAR AND MCAR

For all baselines, the experimental results with standard deviation under MAR and MNAR are showed in Table 11, respectively.

Table 11: Average AUROC(%) with standard deviation on Adult dataset with MAR and MNAR.

| DI Methods | AD Methods | MAR | | | | MNAR | | | |
|---|---|---|---|---|---|---|---|---|---|
| | | Balanced Split | | Skewed Split | | Balanced Split | | Skewed Split | |
| | | $r = 0.2$ | $r = 0.5$ | $r = 0.2$ | $r = 0.5$ | $r = 0.2$ | $r = 0.5$ | $r = 0.2$ | $r = 0.5$ |
| MissForest | I-Forest (Liu et al., 2008) | 60.54 (0.92) | 61.94 (1.07) | 63.07 (0.72) | 63.17 (1.25) | 60.53 (1.40) | 60.24 (1.05) | 61.35 (1.19) | 61.35 (1.20) |
| | Deep SVDD (Ruff et al., 2018) | 61.53 (6.24) | 56.22 (7.75) | 57.55 (10.03) | 57.00 (7.14) | 54.90 (7.71) | 57.54 (4.33) | 56.27 (6.25) | 58.50 (3.49) |
| | NeutraL AD (Qiu et al., 2021) | 52.29 (1.51) | 51.96 (1.01) | 54.37 (1.42) | 53.32 (0.67) | 53.07 (1.26) | 50.82 (2.35) | 52.97 (1.23) | 52.63 (2.17) |
| MissOT$_{(MLP)}$ | I-Forest (Liu et al., 2008) | 45.63 (2.93) | 41.94 (2.27) | 45.32 (2.31) | 42.44 (0.71) | 44.78 (2.68) | 38.62 (1.52) | 44.53 (1.92) | 38.78 (1.11) |
| | Deep SVDD (Ruff et al., 2018) | 51.68 (4.17) | 39.59 (6.95) | 44.49 (6.54) | 50.04 (5.87) | 45.77 (8.47) | 50.29 (6.28) | 51.47 (3.31) | 49.12 (8.13) |
| | NeutraL AD (Qiu et al., 2021) | 52.54 (0.78) | 47.24 (1.96) | 52.32 (0.69) | 44.75 (2.11) | 49.87 (1.07) | 49.38 (1.07) | 49.85 (1.00) | 49.62 (1.36) |
| ImAD (Ours) | | **77.43** (3.42) | **74.61** (2.18) | **80.61** (2.13) | **73.68** (2.10) | **73.73** (3.57) | **72.35** (1.53) | **76.10** (2.04) | **75.58** (2.44) |

