# OpenReview forum: "ImAD: An End-to-End Method for Unsupervised Anomaly Detection in the Presence of Missing Values"
_ICLR.cc/2024/Conference — Submitted to ICLR 2024_

### Official Review · Reviewer_Rb71 · 2023-10-29

**Soundness:** 2 fair
**Presentation:** 3 good
**Contribution:** 2 fair
**Rating:** 5
**Confidence:** 4

**Summary:**

This work addresses the challenge of anomaly detection in data with missing values. The authors propose a novel approach called ImAD that combines data imputation and anomaly detection in a unified framework. The proposed method generates pseudo-abnormal samples from the latent space to mitigate the imputation bias that the traditional two-stage methods tend to exhibit. Experimental results show that it outperforms baseline methods in common scenarios.

**Strengths:**

The study on anomaly detection in datasets with missing values represents an underexplored area within the traditional field of anomaly detection. The recognition of imputation bias in the two-stage methods seems well-founded. The methodology for generating pseudo-anomalies in the latent space appears to be sound.

**Weaknesses:**

- Section 2.2 introduces a few related studies on anomaly detection with incomplete data, but these methods are not included in the experimental evaluation. The results primarily rely on conventional approaches that do not explicitly account for missing data, such as DeepSVDD.
- Experimental scenarios are too limited. Only two datasets are tested, while there exist many other benchmark datasets available for anomaly detection. The choice of two specific missing rates (0.2 and 0.5) may not realistically represent the practical scenarios. Only AUROC results are presented, omitting other essential metrics.
- Although this paper includes some theoretical analysis in section 3.6, the derivations of the inequalities appear to be straightforward in the context of plain neural nets. It remains unclear whether these inequalities provide more valuable insights than those derived from other related methods, especially on practical utility.

**Questions:**

- In Figure 1, OC-SVM outperforms IForest across different missing rates, but OC-SVM is excluded in other experiments. Can you provide a rationale for this selectiveness?
- Can you provide results using additional performance metrics such as AUPRC?
- Justification on the use of the sinkhorn divergence in the loss function will be helpful.
- It would be interesting to see some results on the robustness in relation to the values of r1 and r2.

---

> ### Author Response · Authors · 2023-11-21
> **Response**
>
> **Comment 1**: Section 2.2 introduces a few related studies on anomaly detection with incomplete data, but these methods are not included in the experimental evaluation. The results primarily rely on conventional approaches that do not explicitly account for missing data, such as DeepSVDD.
>
> **Response**: Thanks for your comments. Actually, the related methods ([Zemicheal & Dietterich 2019], [Fan et al. 2022], and [Sarda et al. 2023]) discussed in Section 2.2 are all two-stage methods: perform missing data imputation first and then conduct anomaly detection. A few two-stage methods are considered in our paper. We used two strong missing data imputation methods including MissForest [1] and GAIN [2] and three strong anomaly detection methods including IForest [3], Deep SVDD [4] and NeutraL AD [5]. Our end-to-end method outperformed all two-stage methods.
>
> References:
>
> [1] Stekhoven, Daniel J., and Peter Bühlmann. "MissForest—non-parametric missing value imputation for mixed-type data." Bioinformatics 28.1 (2012): 112-118.
>
> [2] Yoon, Jinsung, James Jordon, and Mihaela Schaar. "Gain: Missing data imputation using generative adversarial nets." International conference on machine learning. PMLR, 2018.
>
> [3] Liu, Fei Tony, Kai Ming Ting, and Zhi-Hua Zhou. "Isolation forest." 2008 eighth ieee international conference on data mining. IEEE, 2008.
>
> [4] Ruff, Lukas, et al. "Deep one-class classification." International conference on machine learning. PMLR, 2018.
>
> [5] Qiu, Chen, et al. "Neural transformation learning for deep anomaly detection beyond images." International Conference on Machine Learning. PMLR, 2021.
>
> **Comment 2**: Experimental scenarios are too limited. Only two datasets are tested, while there exist many other benchmark datasets available for anomaly detection. The choice of two specific missing rates (0.2 and 0.5) may not realistically represent the practical scenarios. Only AUROC results are presented, omitting other essential metrics.
>
> **Response**: We conducted experiment on **five** more publicly-available real-world datasets and added AUPRC as a new evaluation metric. For the selection of missing rates (0.2 and 0.5), we followed the previous data imputation works [1,2,3].
>
> References:
>
> [1] Yoon, Jinsung, James Jordon, and Mihaela Schaar. "Gain: Missing data imputation using generative adversarial nets." International conference on machine learning. PMLR, 2018.
>
> [2] Camino, Ramiro D., Christian A. Hammerschmidt, and Radu State. "Improving missing data imputation with deep generative models." arXiv preprint arXiv:1902.10666 (2019).
>
> [3] Li, Yuxuan, Ayse Dogan, and Chenang Liu. "Ensemble Generative Adversarial Imputation Network with Selective Multi-Generator (ESM-GAIN) for Missing Data Imputation." 2022 IEEE 18th International Conference on Automation Science and Engineering (CASE). IEEE, 2022.
>
> **Comment 3**: Although this paper includes some theoretical analysis in section 3.6, the derivations of the inequalities appear to be straightforward in the context of plain neural nets. It remains unclear whether these inequalities provide more valuable insights than those derived from other related methods, especially on practical utility.
>
> **Response**: Please take a look at Figure 4 of our latest manuscript. In Figure 4, we visualize the projection results of Botnet dataset in 2-D space. It can be observed that the majority of normal training and testing samples are mapped into the target distribution while most abnormal samples fall outside of the decision boundary. This demonstrates that our method is practical and Theorem 3.1 is effective for real-world scenarios.
>
> **Question 1**: In Figure 1, OC-SVM outperforms IForest across different missing rates, but OC-SVM is excluded in other experiments. Can you provide a rationale for this selectiveness?
>
> **Response**:  First of all, in our main experiments, our method achieves better detection performance than the strong baselines Deep SVDD [1] and NeutraL AD [2] and both of them outperformed OC-SVM in their previous works. Therefore, OC-SVM is not a necessary baseline for our work. The second reason is that, our experiments include diverse experimental settings including different missing rates, three missing mechanisms, different data splitting, and different combinations of imputation and detection methods and OC-SVM has quadratic time and space complexities (very slow on large datasets such as KDD). Therefore, we do not include OC-SVM in our experiments.
>
> References:
>
> [1] Ruff, Lukas, et al. "Deep one-class classification." International conference on machine learning. PMLR, 2018.
>
> [2] Qiu, Chen, et al. "Neural transformation learning for deep anomaly detection beyond images." International Conference on Machine Learning. PMLR, 2021.
>
> **Question 2**: Can you provide results using additional performance metrics such as AUPRC?
>
> **Response**: Yes, we add AUPRC as new metric on all seven datasets and related results are provided in our overall response.

---

> > ### Author Response · Authors · 2023-11-21
> > **Response**
> >
> > **Question3**: Justification on the use of the sinkhorn divergence in the loss function will be helpful.
> >
> > **Response**: In optimal transport theory, due to the high computational cost of Wasserstein distance, Sinkhorn divergence is proposed to approximate it by regularizing the original problem with an entropy term. Therefore, the Sinkhorn (Cuturi 2013) divergence is frequently used for measuring the difference between two probability distributions and has many successful applications in machine learning tasks [1,2,3,4]. In our method, we need to project unknown data distribution into a target distribution and Sinkhorn divergence is one of the best choices. By the way, Figure 4 in our latest manuscript shows that our method indeed mapped the normal training data to the target distribution, which demonstrates the effectiveness of sinkhorn divergence.
> >
> > References:
> >
> > [1] Cuturi, Marco. "Sinkhorn distances: Lightspeed computation of optimal transport." Advances in neural information processing systems 26 (2013).
> >
> > [2] Altschuler, Jason, Jonathan Niles-Weed, and Philippe Rigollet. "Near-linear time approximation algorithms for optimal transport via Sinkhorn iteration." Advances in neural information processing systems 30 (2017).
> >
> > [3] Genevay, Aude, Gabriel Peyré, and Marco Cuturi. "Learning generative models with sinkhorn divergences." International Conference on Artificial Intelligence and Statistics. PMLR, 2018.
> >
> > [4] Muzellec, Boris, et al. "Missing data imputation using optimal transport." International Conference on Machine Learning. PMLR, 2020.
> >
> > **Question 4**: It would be interesting to see some results on the robustness in relation to the values of r1 and r2.
> >
> > **Response**: For your concern on radii $r_1, r_2$, we conduct related experiments on Arrhythmia dataset by changing $r_1, r_2$ and experimental details and results are provided in the following tables.
> >
> > As showed in the following table, we change the dimension $d$ of latent space and then get $r = \sigma \sqrt{F^{-1}_{d}(p)}$ (See Proposition A.1).
> >
> > Simultaneously we keep consistent with the experiment setting in our manuscripts (See Appendix F) and get
> > \begin{equation}
> > r_1 = 0.5 \sqrt{F^{-1}_{d}(0.9)}
> > \end{equation}
> >
> > \begin{equation}
> > r_2 = \sqrt{F^{-1}_{d}(0.9)}
> > \end{equation}
> > \begin{matrix}
> >     \hline
> >     Radius & & & Latent Dimension (d) & &  \\\\
> > \hline
> >     & 16 & 32 & 64 & 128 & 256 \\\\
> >     \hline
> >   r_1 & 2.42 & 3.26 & 4.44 & 6.10 & 8.45 \\\\
> >     \hline
> >     r_2 & 4.85 & 6.52 & 8.88 & 12.20 & 16.90 \\\\
> > \hline
> > \end{matrix}
> >
> > The following table shows the average AUROC(\%) and AUPRC(\%) on Arrhythmia dataset when changing the radii $r_1, r_2$. It can be observed that our method is not quite sensitive to the changes of radii $r_1, r_2$ and the performance declines with the decrease of the latent dimension, which is reasonable since there will be more information loss when the latent dimension becomes smaller.
> > \begin{matrix}
> >     \hline
> >     \text{Radius} & \text{Radius}  & \text{AUROC(\\%)} & \text{AUROC(\\%)} & \text{AUPRC(\\%)} & \text{AUPRC(\\%)}\\\\
> >     \hline
> >     r_1 & r_2& p=0.2 & p=0.5 & p=0.2 & p=0.5 \\\\
> >     \hline
> >     2.42 & 4.85 & 80.83 & 78.59 & 79.75 & 77.51 \\\\
> >     & & (1.91) & (1.34) & (1.74) & (2.39) \\\\
> >     \hline
> >     3.26 & 6.52 & 80.06 & 79.37 & 81.06 & 79.19 \\\\
> >     & & (2.00) & (0.56) & (0.87) & (2.12) \\\\
> >     \hline
> >     4.44 & 8.88 & 81.61 & 80.30 & 80.81 & 80.29\\\\
> >     & & (1.05) & (1.21) & (2.06) & (1.09) \\\\
> >     \hline
> >     6.10 & 12.20 & 82.24 & 81.76 & 83.74 & 83.37\\\\
> >     & & (1.76) & (1.19) & (1.85) & (1.36) \\\\
> >     \hline
> >     8.45 & 16.90 & 81.74 & 79.53 & 81.68 & 80.17\\\\
> >     & & (2.05) & (0.73) & (2.80) & (1.05) \\\\
> >     \hline
> > \end{matrix}

---

### Official Review · Reviewer_8XjW · 2023-11-01

**Soundness:** 2 fair
**Presentation:** 2 fair
**Contribution:** 2 fair
**Rating:** 5
**Confidence:** 4

**Summary:**

In this paper the authors present an anomaly detection framework which can handle missing values. One of the key challenges being addressed is to ensure that the imputation model trained on normal data should generalize to abnormal data. Authors pursue a generative modeling approach to generate pseudo-abnormal samples and learn an imputation model on both samples.  A composite loss term evaluating imputation, anomaly and reconstruction losses is formulated for the proposed method. Authors provide results on sample UCI datasets and compare performance to deep learning based anomaly detection and other AD baselines.

**Strengths:**

1) Paper is technically sound and the presentation is good.
2) Proposed framework is simple and makes sense mostly. The problem although heavily studied, there is not a lot of literature on AD with missing data.
3) Empirical results compare performance against both deep AD and other regular AD baselines.

**Weaknesses:**

1) Figure 1 uses AUROC on y-axis whereas for anomaly detection I think AUPR is better to motivate as most anomaly detection techniques have poor precision. Also Adult and KDD are UCI datasets which do not have a lot of relevance today as most techniques perform superlatively on these datsets. Given that the framework is applicable for missing values, it makes more sense to benchmark/motivate on real world use cases for anomaly detection (IoT sensor data, etc)
2) Section 3.5 seems way too short and not very insightful with the purpose of just being added in the paper to have such a section in the main paper.
3) Also on a more real-world applicability note, imputation is not often used within industry and missing data if any is more often discarded. I would encourage authors to think of very strong real-world insights of when imputation should be used and when should it be avoided.

**Questions:**

1) I would like to see stronger results on AUPR for high dimensional real-world datasets which actually have a lot of missing values. Some real-world datasets from IoT sensor domains or others will definitely help in improving contributions of this paper. Some other questions are mentioned in the weakness section above

---

> ### Author Response · Authors · 2023-11-21
> **Response**
>
> **Comment 1**: Figure 1 uses AUROC on y-axis whereas for anomaly detection I think AUPR is better to motivate as most anomaly detection techniques have poor precision. Also Adult and KDD are UCI datasets which do not have a lot of relevance today as most techniques perform superlatively on these datsets. Given that the framework is applicable for missing values, it makes more sense to benchmark/motivate on real world use cases for anomaly detection (IoT sensor data, etc)
>
> **Response**: Following your concern, we conduct experiments on **five** new real-world datasets and add AUPRC as a new evaluation metric. Detailed data information and experimental results can be found in our overall response.  The KDD and Adult datasets are important benchmarks frequently used in previous anomaly detection works [1,2,3,4,5] and we hence included them to improve the persuasiveness.
>
> References:
>
> [1] Zong, Bo, et al. "Deep autoencoding gaussian mixture model for unsupervised anomaly detection." International conference on learning representations. 2018.
>
> [2] Lahoti, Preethi, et al. "Fairness without demographics through adversarially reweighted learning." Advances in neural information processing systems 33 (2020): 728-740.
>
> [3] Qiu, Chen, et al. "Neural transformation learning for deep anomaly detection beyond images." International Conference on Machine Learning. PMLR, 2021.
>
> [4] Buyl, Maarten, and Tijl De Bie. "Optimal transport of classifiers to fairness." Advances in Neural Information Processing Systems 35 (2022): 33728-33740.
>
> [5] Han, Xiao, et al. "Achieving Counterfactual Fairness for Anomaly Detection." Pacific-Asia Conference on Knowledge Discovery and Data Mining. Cham: Springer Nature Switzerland, 2023.
>
> **Comment 2**: Section 3.5 seems way too short and not very insightful with the purpose of just being added in the paper to have such a section in the main paper.
>
> **Response**: For target distribution
> $
> \mathcal{D}_{\mathbf{z}} \sim \mathcal{N}(\mathbf{0}, \sigma^2\mathbf{I}_d)
> $,
> we need to a constrained sampling radius $r$ to get corresponding truncated Gaussian distribution. Section 3.5 gives the lower bound of the constrained sampling radius when given a sampling probability $p$.
>
> **Comment 3**: Also on a more real-world applicability note, imputation is not often used within industry and missing data if any is more often discarded. I would encourage authors to think of very strong real-world insights of when imputation should be used and when should it be avoided.
>
> **Response**:  In fact, in many real-world scenarios, data imputation is necessary and crucial. For instance, in rare disease diagnosis and fault detection of sophisticated equipments, we cannot discard the incomplete samples and we have to make decisions based on the incomplete samples in time. In recommendation systems especially collaborative filtering problems, samples (e.g. users) with missing values commonly cannot be thrown away. In chemical processes (e.g. [1]), we need to conduct fault detection for each sample to ensure product quality and system safety. We appreciate your suggestion and enhanced the motivation in the section of Introduction.
>
> Reference:
>
> [1] Fan et al. Kernel-based statistical process monitoring and
> fault detection in the presence of missing data. IEEE Transactions on Industrial Informatics, 18
> (7):4477–4487, 2022.
>
> **Question 1**: I would like to see stronger results on AUPR for high dimensional real-world datasets which actually have a lot of missing values. Some real-world datasets from IoT sensor domains or others will definitely help in improving contributions of this paper. Some other questions are mentioned in the weakness section above
>
> **Response**: Yes, we have added more datasets to the experiments and included AUPRC. Our method outperformed all baselines in almost all cases in terms of both AUPRC and AUROC.

---

### Official Review · Reviewer_sFKw · 2023-11-01

**Soundness:** 3 good
**Presentation:** 3 good
**Contribution:** 3 good
**Rating:** 5
**Confidence:** 3

**Summary:**

The authors propose a new end-to-end approach called ImAD, which integrates data imputation with anomaly detection in a unified optimization problem. ImAD addresses the imputation bias issue and demonstrates improved detection performance on balanced and skewed data compared to existing methods in various missing data scenarios.

**Strengths:**

- The concept of initially generating pseudo-abnormal samples and subsequently constructing an imputation model from both the normal and pseudo-abnormal datasets is a novel approach.
- The authors provide a theoretical analysis of the generation and detection of pseudo-abnormal samples.
- The observed performance enhancement in comparison to the baseline methods is substantial and noteworthy.

**Weaknesses:**

- The current scope of experiments is somewhat limited, and it is strongly recommended to expand the experimental evaluation to include a broader range of datasets. This will facilitate a more comprehensive validation of the efficacy of the proposed method.
- The assumption of a Gaussian distribution in the latent space could potentially impose limitations on the practical applicability of the proposed method. Real-world datasets with inherent complexity may not fit into a single Gaussian distribution, raising concerns about the generalizability of the approach to such intricate data scenarios.

**Questions:**

- How can we ensure that the generated pseudo abnormal samples are similar to real abnormal samples? Because the distribution of pseudo abnormal samples in latent space is assumed, it can differ significantly from that of real abnormal samples. This difference can lead to strange imputation.
- When dealing with a dataset of very high dimensionality, is the proposed method scalable? Will the assumption in generating pseudo-abnormal samples still hold?
- There is an excessive use of similar symbols placed above characters, which can impede readability.
- Typo. The last paragraph of 2 Related Work. impuate -> impute.

---

> ### Author Response · Authors · 2023-11-21
> **Response**
>
> **Comment 1**: The current scope of experiments is somewhat limited, and it is strongly recommended to expand the experimental evaluation to include a broader range of datasets. This will facilitate a more comprehensive validation of the efficacy of the proposed method.
>
> **Response**: Thanks for your well-intentioned suggestion. We added five new real-world datasets to the experiments and one more evaluation metric AUPRC. All experimental results are provided in our overall response and the new experimental results are complemented to section of Experiments in our latest manuscript.
>
> **Comment 2**: The assumption of a Gaussian distribution in the latent space could potentially impose limitations on the practical applicability of the proposed method. Real-world datasets with inherent complexity may not fit into a single Gaussian distribution, raising concerns about the generalizability of the approach to such intricate data scenarios.
>
> **Response**: This is really an insightful comment and we absolutely agree with you. Indeed, the real-world data scenarios could be more complex and we may consider multiple Gaussians.
>
> We also thought about projecting the normal data into multiple truncated Gaussian distributions but there are two difficulties. First, the number of classes in normal data is usually unknown, which makes it difficult to determine the number of Gaussians. Second, the spatial locations of multiple Gaussians or hyperballs are not easy to determine.
>
> On the other hand, if the structure of real data is not very complex, a sufficiently large neural network is able to transform the unknown data distribution to any target distribution such as a single Gaussian, which is similar to the feasibility of GAN (Goodfellow et al. 2014) converting a single Gaussian distribution to any data distribution.
> In unsupervised anomaly detection, the most important thing is to find a reliable decision boundary using the normal samples. Therefore, the target distribution is supposed to be compact, which ensures that normal data would lie in high-density regions in the latent space and we can obtain a reliable decision boundary easily.
>
> We appreciate your comments.
>
> **Question 1**: How can we ensure that the generated pseudo abnormal samples are similar to real abnormal samples? Because the distribution of pseudo abnormal samples in latent space is assumed, it can differ significantly from that of real abnormal samples. This difference can lead to strange imputation.
>
> **Response**: This is a great question. Actually, there is no need to ensure that the generated pseudo-abnormal samples are very similar to real abnormal samples. We only want to ensure that the generated pseudo-abnormal samples closely surround the normal samples in the original data space, which means the generated pseudo-abnormal samples interpolate between the normal region and the abnormal region in the original data space and hence form a meaningful decision boundary. This goal can be achieved by our design of the distributions in the latent space and the Lipschitz-continuity of the neural network. Specifically, as shown by Figure 2 of our paper, in the latent space, the pseudo-abnormal samples surround the normal samples. According to our Theorem 3.1(a), in the original data space, the pseudo-abnormal samples still surround the normal samples.
>
> Please also take a look at Figure 4 of our latest manuscript. We visualize the projection results of Botnet dataset in 2-D space. It can be observed that the majority of normal training and testing samples are mapped into the target distribution while most abnormal samples fall outside of the decision boundary. This demonstrates that our method is practical and Theorem 3.1 is effective for real-world scenarios.
>
> **Question 2** :  When dealing with a dataset of very high dimensionality, is the proposed method scalable? Will the assumption in generating pseudo-abnormal samples still hold?
>
> **Response**: Our method projects the original high-dimensional data into the target distribution, which can reduce the dimensionality significantly, and the anomaly score is defined in the low-dimensional latent space. Therefore, our method is scalable to high-dimensional data. In this revision, the new datasets Segerstolpe and Usoskin especially have high-dimension and our method still outperformed all baselines significantly.
>
> **Question 3**: There is an excessive use of similar symbols placed above characters, which can impede readability.
>
> **Response**: Thanks for pointing it out. We enhanced the related descriptions and made them easy to read and understand.

---

### Author Response · Authors · 2023-11-21
**Overall Response for all Reviewers (Additional Experiments)**

The authors highly appreciate all reviewers' comments and suggestions.

In our original paper, because we considered diverse experimental settings including different missing rates ($p=\{0.2, 0.5\}$), three different missing mechanisms (MCAR, MAR, and MNAR), two different data splittings (balanced and skewed), and different combinations of imputation and detection methods,  we used only two large benchmark datasets of anomaly detection, say Adult and KDD.

Given the consistent concerns of reviewers on experimental evaluation, we added five more real-world and publicly-available datasets to the experiments and one more evaluation metric AUPRC. The five new datasets (Botnet, Arrhythmia, Speech, Segerstolpe, and Usoskin) are from diverse fields such as biology and medicine.

The descriptions about all the seven datasets are provided in the following table.

\begin{matrix}
    \hline
    \text{Dataset} &  \text{Field} & \text{Features} &  \text{Instances} &  \text{Normal Samples} &  \text{Abnormal Samples}\\\\
    \hline
     \text{Adult} &  \text{income census} & 14 & 30,162 & 22,658 & 7,508\\\\
     \text{KDD} &  \text{cybersecurity} & 121 & 494,021 & 396,743 & 97,278\\\\
    \hline
     \text{Botnet(IOT)} &  \text{cybersecurity} & 115 & 40,607 & 13,113 & 27,494\\\\
     \text{Arrhythmia} &  \text{medical diagnosis} & 274 & 452 & 320 & 132\\\\
     \text{Speech} &  \text{speech recognition} & 400 & 3,686 & 3,625 & 61\\\\
     \text{Segerstolpe} &  \text{cell analysis} & 1,000 & 702 & 329 & 372\\\\
 \text{Usoskin} &  \text{cell analysis} & 25,334 & 610 & 232 & 378\\\\
\hline
\end{matrix}
We repeat the experiment of each setting five times and report the average AUROC(\%) and AUPRC(\%) with standard deviation in the following tables. All the experiments are implemented under balanced data splitting. Furthermore, we also complement the experimental results on Adult and KDD with metric AUPRC in the following tables.

It should be pointed out that our method outperformed the baselines in almost all cases significantly and all the additional experimental results are provided in the following tables.

**AUROC and AUPRC (\%, mean and std) on Arrhythmia dataset with MCAR.**
\begin{matrix}
   \hline
   \text{DI Methods} & \text{AD Methods} & \text{AUROC(\\%)} &  \text{AUROC(\\%)} & \text{AUPRC(\\%)} & \text{AUPRC(\\%)}  \\\\
    & & r = 0.2 & r = 0.5 & r = 0.2 & r = 0.5 \\\\
\hline
    &\text{I-Forest} & 80.72 & 81.54 & 77.91 & 77.95 \\\\
    & & (0.62) & (0.95) & (1.85) & (0.97)\\\\
    \text{MissForest}&\text{Deep SVDD} & 72.63 & 7ne5.80 & 70.94 & 77.39 \\\\
    & & (0.99) & (4.07) & (0.75) & (4.55)\\\\
    &\text{NeutraL AD} & 47.38  & 44.30 & 50.87 & 50.12 \\\\
    & & (4.81) & (2.11) & (3.53) & (2.52)\\\\
\hline
&\text{I-Forest} & 77.19 & 76.29 & 76.40 & 76.29 \\\\
     & & (0.81) & (1.35) & (1.86) & (1.35)\\\\
    \text{GAIN}&\text{Deep SVDD} & 57.14 & 48.86 & 59.35 & 54.03 \\\\
   & & (5.41) & (2.35) & (2.58) & (2.45)\\\\
    &\text{NeutraL AD} & 37.96 & 33.98 & 42.57 & 42.35 \\\\
    & & (5.09) & (4.12) & (2.56) & (1.96)\\\\
    \hline
\text{ImAD (Ours)} & & \textbf{82.24} & \textbf{81.76} & \textbf{83.74} & \textbf{83.37}\\\\
& & (1.76) & (1.19) & (1.85) & (1.36) \\\\
    \hline
\end{matrix}

**AUROC, AUPRC (\%, mean and std) on Speech dataset with MCAR.**
\begin{matrix}
   \hline
   \text{DI Methods} & \text{AD Methods} & \text{AUROC(\\%)} &  \text{AUROC(\\%)} & \text{AUPRC(\\%)} & \text{AUPRC(\\%)}  \\\\
    & & r = 0.2 & r = 0.5 & r = 0.2 & r = 0.5 \\\\
\hline
     & \text{I-Forest} & 28.58 & 29.09 & 36.83 & 37.29 \\\\
    & & (2.95) & (1.14) & (1.06) & (0.76)\\\\
    \text{MissForest}& \text{Deep SVDD} & 60.37 & 40.14 & 58.93 & 42.08 \\\\
    & & (0.87) & (4.30) & (1.35) & (2.16)\\\\
    & \text{NeutraL AD} & 56.51  & 54.11 & 55.44 & 52.26 \\\\
    & & (4.87) & (3.77) & (4.36) & (3.97)\\\\
    \hline
     & \text{I-Forest} & 29.33 & 29.23 & 39.92 & 40.04 \\\\
     & & (0.59) & (2.13) & (0.21) & (0.63)\\\\
     \text{GAIN}& \text{Deep SVDD} & 54.95 & 46.54 & 54.38 & 47.54 \\\\
    & & (1.79) & (2.10) & (0.96) & (1.75)\\\\
    & \text{NeutraL AD} & 56.80 & 57.24 & 54.76 & 55.05 \\\\
    & & (4.89) & (5.51) & (4.58) & (5.58)\\\\
    \hline
\text{ImAD (Ours)} & & \textbf{61.94}& \textbf{58.66} & \textbf{60.43} & \textbf{58.13}\\\\
& &  (2.77) & (1.40) & (3.33) & (1.48) \\\\
\hline
\end{matrix}

---

> ### Author Response · Authors · 2023-11-21
> **Overall Response for all Reviewers (Additional Experiments)**
>
> **AUROC, AUPRC (\%, mean and std) on Segerstolpe dataset with MCAR.**
> \begin{matrix}
>    \hline
>    \text{DI Methods} & \text{AD Methods} & \text{AUROC(\\%)} &  \text{AUROC(\\%)} & \text{AUPRC(\\%)} & \text{AUPRC(\\%)}  \\\\
>     & & r = 0.2 & r = 0.5 & r = 0.2 & r = 0.5 \\\\
> \hline
>      & \text{I-Forest} & 94.91 & 96.68 & 95.94 & \textbf{97.56} \\\\
>     & & (1.35) & (0.79) & (1.23) & (0.59)\\\\
>     \text{MissForest}& \text{Deep SVDD} & 96.20 & 89.24 & 97.53 & 90.65 \\\\
>     & & (2.66) & (1.44) & (1.40) & (0.57)\\\\
>     & \text{NeutraL AD} & 97.89 & 89.38 & 97.71 & 84.61 \\\\
>     & & (1.45) & (2.80) & (1.76) & (3.78)\\\\
>     \hline
>      & \text{I-Forest} & 94.25 & 92.07 & 96.14 & 93.94 \\\\
>      & & (0.90) & (1.82) & (0.75) & (1.62)\\\\
>     \text{GAIN} & \text{Deep SVDD} & 92.46 & 94.32 & 92.25 & 92.88 \\\\
>     & & (4.25) & (1.93) & (2.40) & (1.26)\\\\
>     & \text{NeutraL AD} & 97.52 & 90.10 & 97.52 & 90.10 \\\\
>     & & (0.37) & (0.90) & (1.02) & (0.82)\\\\
>     \hline
> \text{ImAD (Ours)} & & \textbf{99.14} & \textbf{96.86} & \textbf{98.98} & 96.85 \\\\
> & & (0.88) & (0.67) & (1.18) & (0.54) \\\\
>     \hline
> \end{matrix}
>
> **AUROC, AUPRC (\%, mean and std) on Usoskin dataset with MCAR.**
> \begin{matrix}
> \hline
>    \text{DI Methods} & \text{AD Methods} & \text{AUROC(\\%)} &  \text{AUROC(\\%)} & \text{AUPRC(\\%)} & \text{AUPRC(\\%)}  \\\\
>     & & r = 0.2 & r = 0.5 & r = 0.2 & r = 0.5 \\\\
> \hline
>      & \text{I-Forest} & 45.19 & 49.64 & 46.97 & 49.74 \\\\
>     & & (4.56) & (7.43) & (3.04) & (5.64)\\\\
>     \text{MissForest}& \text{Deep SVDD} & 37.47 & 43.61 & 50.55 & 55.05 \\\\
>     & & (3.83) & (7.49) & (2.02) & (4.81)\\\\
>     & \text{NeutraL AD} & 57.43 & 53.74 & 63.65 & 61.05 \\\\
>     & & (4.59) & (2.27) & (2.40) & (4.16)\\\\
>     \hline
>     & \text{I-Forest} & 40.96 & 37.11 & 46.29 & 42.86 \\\\
>      & & (2.02) & (2.12) & (1.76) & (1.22)\\\\
>     \text{GAIN} & \text{Deep SVDD} & 49.99 & 65.48 & 54.85 & 64.54 \\\\
>     & & (5.69) & (2.94) & (1.61) & (0.74)\\\\
>     & \text{NeutraL AD} & 56.18 & 64.80 & 64.85 & 73.33 \\\\
>     & & (2.62) & (1.85) & (2.68) & (1.31)\\\\
>     \hline
>     \text{ImAD (Ours)} & & \textbf{84.95}  & \textbf{79.23} & \textbf{85.48}& \textbf{80.06} \\\\
>     & &(1.29) & (2.49) & (2.34) & (3.40) \\\\
> \hline
> \end{matrix}
>
> **AUROC, AUPRC (\%, mean and std) on Botnet dataset with MCAR.**
> \begin{matrix}
>     \hline
>     \text{DI Methods} & \text{AD Methods} & \text{AUROC(\\%)} &  \text{AUROC(\\%)} & \text{AUPRC(\\%)} & \text{AUPRC(\\%)}  \\\\
>     & & r = 0.2 & r = 0.5 & r = 0.2 & r = 0.5 \\\\
>     \hline
>     & \text{I-Forest} & 95.72 & 93.86 & 97.25 & 95.68 \\\\
>     & & (0.96) & (0.70) & (0.69) & (0.52)\\\\
>     \text{MissForest} & \text{Deep SVDD} & 96.72 & 97.51 & 96.60 & 97.62 \\\\
>     & & (0.87) & (0.94) & (0.80) & (0.89)\\\\
>     & \text{NeutraL AD} & 99.04 & 97.27 & 98.92 & 97.68 \\\\
>     & & (0.26) & (0.59) & (0.24) & (0.53)\\\\
>     \hline
>     & \text{I-Forest} & 96.16 & 94.01 & 97.61 & 96.18 \\\\
>      & & (0.24) & (0.73) & (0.21) & (0.44)\\\\
>      \text{GAIN} & \text{Deep SVDD} & 98.68 & 98.02 & 98.35 & 97.59 \\\\
>     & & (0.11) & (0.41) & (0.14) & (0.46)\\\\
>     & \text{NeutraL AD} & 97.42 & \textbf{99.56} & 96.89 & 99.41 \\\\
>     & & (0.33) & (0.27) & (0.36) & (0.35)\\\\
>     \hline
>     \text{ImAD (Ours)} & & \textbf{99.71} & 99.53 & \textbf{99.68} & \textbf{99.58} \\\\
>     & & (0.22)& (0.25)&(0.24) &(0.20) \\\\
>     \hline
> \end{matrix}
>
> **AUPRC (\%, mean and std) on KDD and Adult dataset with MCAR.**
> \begin{matrix}
>     \hline
>     \text{DI Methods} & \text{AD Methods} & \text{KDD} & \text{KDD} & \text{Adult} & \text{Adult} \\\\
>     & & r = 0.2 & r = 0.5 & r = 0.2 & r = 0.5 \\\\
>     \hline
>     & \text{I-Forest} & 93.24 & 93.21 & 57.12 & 56.80 \\\\
>     & & (2.38) & (1.92) & (2.16) & (1.27)\\\\
>     \text{MissForest} & \text{Deep SVDD} & 85.77 & 88.79 & 55.31 & 55.45 \\\\
>     & & (2.95) & (1.29) & (2.91) & (1.72)\\\\
>     & \text{NeutraL AD} & 93.87 & \textbf{94.88} & 50.07 & 52.27 \\\\
>     & & (1.57) & (2.86) & (6.50) & (3.61)\\\\
>     \hline
>     & \text{I-Forest} & 90.33 & 89.52 & 57.05 & 56.87 \\\\
>      & & (1.58) & (1.07) & (1.02) & (1.09)\\\\
>     \text{GAIN} & \text{Deep SVDD} & 88.36 & 85.45 & 57.61 & 59.55 \\\\
>     & & (3.52) & (2.77) & (4.24) & (2.34)\\\\
>     & \text{NeutraL AD} & 84.61 & 84.08 & 53.00 & 59.06 \\\\
>     & & (1.30) & (1.71) & (6.80) & (3.97)\\\\
>     \hline
>     \text{ImAD (Ours)} & & \textbf{95.96} & 91.58 & \textbf{73.42} & \textbf{71.50} \\\\
>     & &(0.18) & (0.32)&(2.08) &(2.02)\\\\
>     \hline
> \end{matrix}
> We added the experimental results to section of Experiments in our latest manuscript and highlighted specific changes in blue.

---

### Meta-Review · Area_Chair_giHD · 2023-12-04

**Metareview:**

The reviewers were unanimous in their vote to reject, feeling that the submission was not ready for publication. Issues raised included lack of meaningful interpretation of experimental results.

**Justification For Why Not Higher Score:**

Reviewers do not support acceptance.

**Justification For Why Not Lower Score:**

N/A

---

### Decision · Program_Chairs · 2024-01-16

Reject